# Activation of individual L1 retrotransposon instances is restricted to cell-type dependent permissive loci

Claude Philippe[1,2,3], Dulce B Vargas-Landin[1,2,4†], Aurélien J Doucet[1,2,3], Dominic van Essen[1,2], Jorge Vera-Otarola[1,2‡], Monika Kuciak[1,2,5§], Antoine Corbin[5¶], Pilvi Nigumann[1,2], Gaël Cristofari[1,2,3*]

[1]INSERM U1081, CNRS UMR 7284, Institute for Research on Cancer and Aging of Nice, Nice, France; [2]Faculty of Medicine, University of Nice-Sophia Antipolis, Nice, France; [3]FHU OncoAge, University of Nice-Sophia Antipolis, Nice, France; [4]Ecole Normale Supérieure, Paris, France; [5]Ecole Normale Supérieure de Lyon, Lyon, France

**\*For correspondence:** Gael. Cristofari@unice.fr

**Present address:** [†]Harry Perkins Institute of Medical Research, The University of Western Australia, Perth, Australia; [‡]Escuela de Medicina, Pontificia Universidad Católica de Chile, Santiago, Chile; [§]Department of Genetic Medicine and Development, University of Geneva Medical School, Geneva, Switzerland; [¶]CIRI, International Center for Infectiology Research, Ecole Normale Supérieure de Lyon, Lyon, France

**Abstract** LINE-1 (L1) retrotransposons represent approximately one sixth of the human genome, but only the human-specific L1HS-Ta subfamily acts as an endogenous mutagen in modern humans, reshaping both somatic and germline genomes. Due to their high levels of sequence identity and the existence of many polymorphic insertions absent from the reference genome, the transcriptional activation of individual genomic L1HS-Ta copies remains poorly understood. Here we comprehensively mapped fixed and polymorphic L1HS-Ta copies in 12 commonly-used somatic cell lines, and identified transcriptional and epigenetic signatures allowing the unambiguous identification of active L1HS-Ta copies in their genomic context. Strikingly, only a very restricted subset of L1HS-Ta loci - some being polymorphic among individuals - significantly contributes to the bulk of L1 expression, and these loci are differentially regulated among distinct cell lines. Thus, our data support a local model of L1 transcriptional activation in somatic cells, governed by individual-, locus-, and cell-type-specific determinants.

## Introduction

At least half of our DNA is derived from repeated and dispersed sequences called retrotransposons, a class of mobile genetic elements which proliferate via an RNA-mediated copy-and-paste mechanism termed retrotransposition (see [*Burns and Boeke, 2012*; *Hancks and Kazazian, 2012*; *Richardson et al., 2015*] for recent reviews). Since most copies have accumulated mutations or are truncated, they are unable to initiate new cycles of retrotransposition and could be considered as molecular fossils (although in some cases they might nevertheless still be transcriptionally active [*Macia et al., 2011*]). In contrast, the youngest and human-specific L1 subfamily, 'transcribed L1, subset a' or L1HS-Ta, continues to retrotranspose and to accumulate in modern human genomes (*Boissinot et al., 2000*). Hence, each individual has hundreds of additional copies not present in the reference genome, referred to as 'non-reference' L1HS-Ta, which contribute to our genetic diversity (*Xing et al., 2009*; *Cordaux and Batzer, 2009*; *Ewing and Kazazian, 2010*; *Beck et al., 2010*; *Huang et al., 2010*; *Iskow et al., 2010*; *Kidd et al., 2010*; *Lupski, 2010*; *Ewing and Kazazian, 2011*; *Ray and Batzer, 2011*; *Stewart et al., 2011*; *Mir et al., 2015*). Recent advances in deep-sequencing technologies have also led to the discovery that L1HS-Ta are not only able to mobilize in the germline, in the early embryo and in embryonic stem cells - resulting in inheritable genetic variations (*van den Hurk et al., 2007*; *Wissing et al., 2012*; *Hancks and Kazazian, 2012*; *Macia et al.,*

**eLife digest** Retrotransposons, also known as jumping genes, have invaded the genomes of most living organisms. These fragments of DNA have the ability to move or copy themselves from one location of a chromosome to another. Depending on where they insert themselves, retrotransposons can modify the sequence of nearby genes, which can alter or even abolish their activity. Although these genetic modifications have contributed significantly to the evolution of different species, retrotransposons can also have detrimental effects; for example, by causing new cases of genetic diseases.

Adult human cells have a number of mechanisms that work to keep the activity of retrotransposons at a very low level. However, in many types of cancers retrotransposons escape these defense mechanisms and 'jump' actively. This is thought to contribute to the development and spread of cancerous tumors.

To understand how jumping genes are mobilized, a fundamental question must be answered: is the high jumping gene activity observed in some cell types a result of activating many copies of the retrotransposons, or only a few of them? This question has been difficult to address because there are more than one hundred copies of retrotransposons that could potentially move in humans, many of which have not even been referenced in the human genome map. Furthermore, each copy is almost identical to another one, making it difficult to discriminate between them.

Philippe et al. have now developed an approach that can map where individual retrotransposons are located in the genome of normal and cancerous cells and measure how active these jumping genes are. This revealed that only a very restricted number of them are active in any given cell type. Moreover, different subsets of jumping genes are active in different cell types, and their locations in the genome often do not overlap.

Thus, whether jumping genes are activated depends on the cell type and their position in the genome. These results are in contrast to the prevalent view that retrotransposons are activated in a more widespread manner across the genome, at least in cancerous cells. Overall, Philippe et al.'s findings pave the way towards characterizing the chromosome regions in which retrotransposons are frequently activated and understanding how they contribute to cancer and other diseases.

2014) - but can also retrotranspose in some somatic tissues such as brain (*Faulkner et al., 2009*; *Coufal et al., 2009*; *Baillie et al., 2011*; *Evrony et al., 2012*; *Erwin et al., 2014*; *Richardson et al., 2014*; *Upton et al., 2015*), and in many epithelial cancers (*Miki et al., 1992*; *Iskow et al., 2010*; *Solyom et al., 2012*; *Shukla et al., 2013*; *Rodić and Burns, 2013*; *Pitkänen et al., 2014*; *Helman et al., 2014*; *Tubio et al., 2014*; *Goodier, 2014*; *Ewing et al., 2015*; *Rodić et al., 2015*; *Doucet-O'Hare et al., 2015*; *Paterson et al., 2015*).

The overall ability of L1 elements to retrotranspose presumably results from the balance between the activities of the L1 sequences themselves, and the effects of restricting cellular pathways. The first step required to initiate retrotransposition of a particular L1 instance is its transcriptional activation: this is primarily driven by an internal promoter located within the L1 5' UTR (*Swergold, 1990*; *Minakami et al., 1992*; *Tchénio et al., 2000*; *Athanikar et al., 2004*), but can be repressed by CpG methylation (*Yoder et al., 1997*; *Bourc'his and Bestor, 2004*; *Muotri et al., 2010*; *Wissing et al., 2012*; *Castro-Diaz et al., 2014*). Production of L1 RNA transcripts is essential both for the translation of L1-encoded proteins, ORF1p and ORF2p, which are required for retrotransposition (*Moran et al., 1996*), and to act as a template for reverse transcription itself (*Wei et al., 2001*). After reverse transcription and genomic integration, the sequences of each L1 element can accumulate genetic alterations (mutations, deletions, insertion of nested transposable elements), and these can alter the intrinsic integrity and biochemical activity of these copies. As a result, only a fraction of L1 elements are retrotransposition-competent, even when cloned in a plasmid and tested in cellular assays, with their expression driven by a strong constitutive promoter (*Brouha et al., 2003*; *Beck et al., 2010*). These so-called 'hot' L1 elements are highly enriched among the youngest L1HS-Ta insertions, which are polymorphic among individuals (*Beck et al., 2010*; *Lupski, 2010*; *Beck et al., 2011*). Finally, additional cellular pathways and restriction factors can limit L1 activities

at multiple other stages of the L1 retrotransposition cycle (see [*Heras et al., 2014*; *Richardson et al., 2015*; *Pizarro and Cristofari, 2016*] for reviews).

Our understanding of L1 transcriptional activation, particularly in the context of different cell types, remains extremely limited. Indeed, studying this process is complicated by the extent of L1 insertional polymorphisms in individual genomes and the extreme level of sequence identity between the copies of the youngest (and most-active) L1HS-Ta subfamily and with older copies of retrotransposition-incompetent subfamilies. Theoretically, the high L1 activity observed in particular cell types could result from global unleashing of most L1HS-Ta copies. Alternatively, it could derive from a few deregulated L1HS-Ta instances. To resolve these competing models, we mapped the location of each L1HS-Ta element dispersed in the genome of a panel of normal and transformed human cells, identified a genomic signature for the transcriptionally active copies and investigated the contribution of each of them to the bulk of L1HS-Ta transcripts. We found that individual L1 instances exhibit both locus- and cell-type-specific activation, implying that L1 mutagenic activity originates from 'hot L1' inserted in permissive loci and suggesting an unforeseen new layer of cell-type specific regulation to control endogenous retrotransposons.

## Results

### Global expression of recent L1 elements is variable in human cells

Human L1-derived RNA-transcripts and proteins, required for L1 retrotransposition, are detected in embryonic stem cells, in embryonal carcinoma cells, and other transformed cells or tumors, as well as in neuronal progenitor cells, but not in most primary cells, such as fibroblasts (*Faulkner et al., 2009*; *Coufal et al., 2009*; *Belancio et al., 2010*; *Wissing et al., 2012*). To study the polymorphism and expression of the L1HS-Ta subfamily at the level of individual genomic instances, we selected twelve widely used cell lines belonging to each of these different categories (*Supplementary file 1*). These included 10 cell lines which have been characterized in depth as part of the ENCODE project (*Bernstein et al., 2012*), together with two others – the commonly used embryonic lung fibroblast line MRC-5, and the embryonic carcinoma cell line 2102Ep, which is known to express high levels of endogenous L1HS-Ta (*Leibold et al., 1990*).

As a first estimate of L1 activity, we quantified and compared the endogenous levels of the L1-encoded ORF1p protein in distinct cell-lines. We detected ORF1p expression in whole cell extracts of half of the transformed cell lines (*Figure 1—figure supplement 1*), consistent with the proportion of human tumors expressing ORF1p (*Rodić et al., 2014*) and with previous work (*Belancio et al., 2010*). As expected, no ORF1p was detected in primary fibroblasts. ORF1p associates with the L1 mRNA, and L1 ORF2p, to form a ribonucleoprotein particle (RNP), which mediates the retrotransposition reaction (*Kulpa and Moran, 2006*; *Doucet et al., 2010*). To ensure the highest sensitivity and to enrich for functional ORF1p (at least able to bind RNA), we prepared L1 RNPs by sucrose cushion ultracentrifugation and probed ORF1p by immunoblot. We observed similar results as in whole cell extracts, except that ORF1p was faintly detected in two additional transformed cell lines (HeLa S3, Hep G2, *Figure 1a*). In a complementary approach, we estimated the proportion of L1 transcripts originating from the L1HS-Ta subfamily by counting RNA-seq reads mapped on the L1HS consensus sequence, which encompass subfamily-diagnostic SNPs in the L1 3' UTR sequence (ACA for L1HS-Ta, ACG for L1HS-PreTa and GAG for L1PA2 and older) (*Boissinot et al., 2000*). For this analysis, publicly available data from the hESC line H1 were also included (no stranded polyA+ RNA-seq data were available for MRC-5 and HEK-293 cells). Consistent with L1 RNP quantification, MCF7 and 2102Ep cells exhibit the highest levels of L1HS-Ta RNA-seq tags. Most other transformed cells have intermediate levels, while HCT 116 and primary fibroblasts (BJ, IMR-90) have extremely reduced L1HS-Ta levels (*Figure 1b*). In agreement with previous studies on other hESC, H1 cells express relatively high levels of L1HS-Ta (*Garcia-Perez et al., 2007*; *Macia et al., 2011*). Altogether these data indicate that, in several - but not all - transformed cells and in hESC, L1HS-Ta retrotransposons can escape the epigenetic, transcriptional and post-transcriptional controls that usually limit their expression in most somatic cells (*Faulkner et al., 2009*).

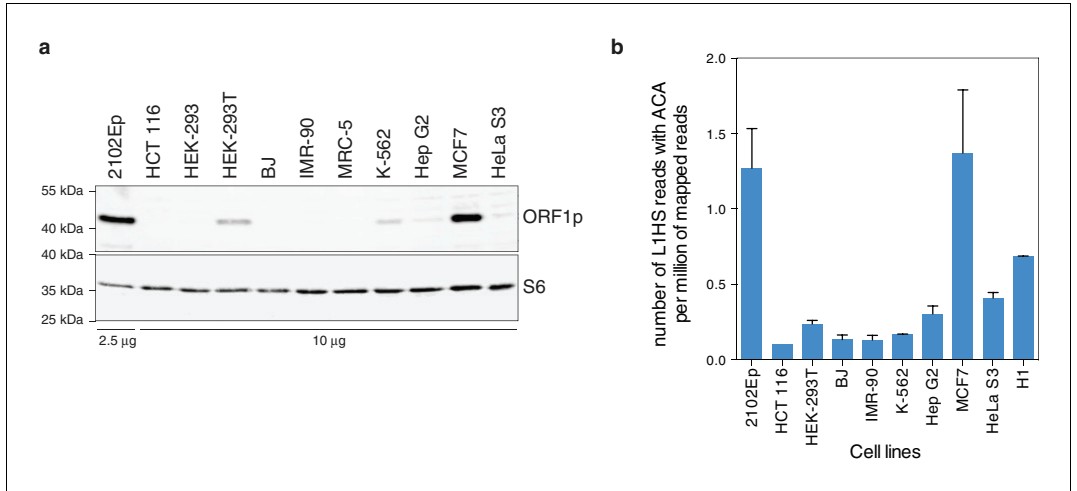

**Figure 1.** Global expression of L1HS elements in a panel of human somatic cell lines. (a) ORF1p immunoblot analysis of L1 RNP accumulation in the indicated cell lines. Top, ORF1p immunoblot. Bottom, S6 Ribosomal Protein immunoblot as loading control. The quantity of RNP loaded is indicated at the bottom of the gel. (b) Global estimate of L1HS-Ta RNA levels obtained by counting RNA-seq reads mapping against the L1HS consensus and containing the Ta-specific ACA diagnostic signature, normalized by the total number of reads mapping in the human reference genome (hg19) (mean ± s.e.m., n=2 except for MCF-7 where n=4, and HCT 116 where n=1). This analysis is based on stranded polyA+ RNA-seq data (*Supplementary file 1*). None were available for MRC-5 and HEK-293 cells, but data obtained from the hESC line H1 were included. See also *Figure 1—figure supplement 1*.

The following figure supplement is available for figure 1:

**Figure supplement 1.** Analysis of L1 ORF1p expression in whole cell extracts of various cell lines by immunoblot.

## A comprehensive map of L1HS-Ta elements in a panel of human cells

We next asked whether the observed expression of L1HS-Ta in some cells results from the transcriptional activation of all or most L1HS-Ta genomic copies, or only of a few of them. As a first step, we mapped the genomic location of all L1HS-Ta elements in each cell line of our panel. To achieve this task, we adapted for deep sequencing an existing method termed ATLAS (*Badge et al., 2003*). In brief, ATLAS-seq relies on the random mechanical fragmentation of the genomic DNA to ensure high-coverage, ligation of adapter sequences, suppression PCR-amplification of L1HS-Ta element junctions, and Ion Torrent sequencing using single-end 400 bp read chemistry (*Figure 2a–c* and *Figure 2—figure supplement 1a*, see also Materials and methods section). A notable aspect of ATLAS-seq is that we combine amplification and mapping of L1 downstream flanking sequence as well as the L1 upstream flanking sequence of full length elements (*Figure 2a–c*). This allows the unambiguous identification of full-length and potentially retrotransposition-competent genomic instances. In total, ATLAS-seq identified 7823 high-confidence L1HS-Ta insertions in the 12 cell lines analyzed, corresponding to 1633 distinct loci and including 358 full length elements (22%) (*Supplementary files 2* and *3*). The human reference genome hg19 contains 485 L1HS-Ta insertions with a detectable 3′ end. On average (± s.d.), each cell line contains 652 (±68) L1HS-Ta copies, including 178 (±12) full-length elements. Among them 393 (±10) are reference insertions, 179 (±18) are non-reference L1HS-Ta previously identified as L1 insertion polymorphisms and catalogued in euL1db (*Mir et al., 2015*), and 80 (±60) are novel insertions (*Figure 2d*). ATLAS-seq recovers 98% of the L1HS-Ta elements previously described as fixed in the human population (see Materials and methods and *Figure 2—figure supplement 1b*), showing that this mapping approach is close to being comprehensive. To further validate the L1HS-Ta elements mapped by ATLAS-seq, we randomly selected and tested by PCR 72 non-reference insertions identified in HEK-293T cells with a broad range of supporting ATLAS-seq reads. Primers could be designed for 70/72 loci. We validated 66/70 of the tested L1HS-Ta, giving a true positive rate of 94% (*Supplementary file 4*). One fifth of

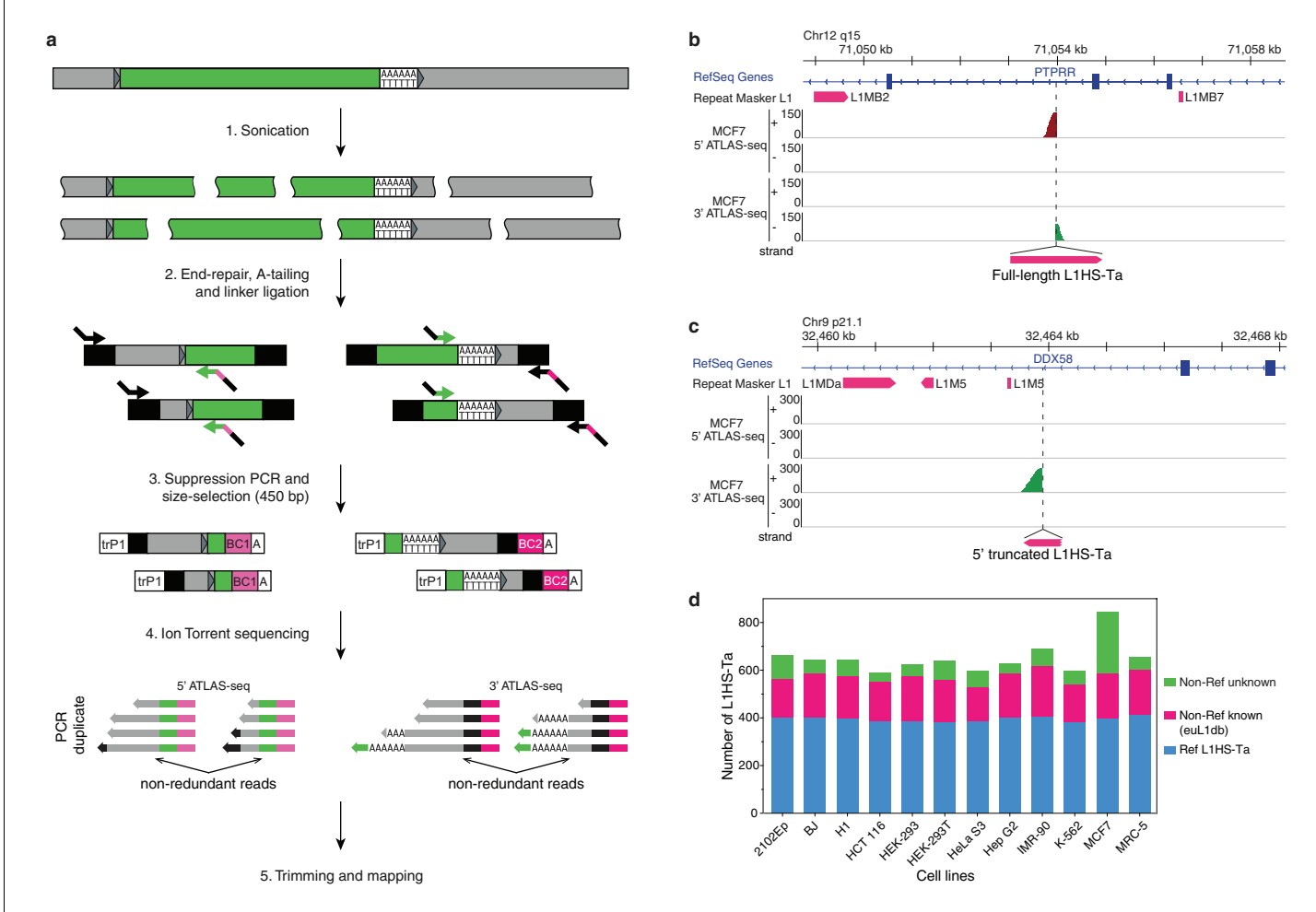

**Figure 2.** The genetic landscape of L1HS-Ta insertional polymorphisms in 12 human somatic cell lines. (a) Principle of the ATLAS-seq procedure. The subsequent in silico steps are described in *Figure 2—figure supplement 1a*. (b–c), Modified IGV genome browser views (*Thorvaldsdóttir et al., 2013*) of two non-reference polymorphic L1 instances detected in MCF7 cells (b, full length L1, note the two adjacent 5'- and 3'-ATLAS-seq peaks; c, truncated L1). (d) L1HS-Ta insertions found in the various cells of the studied panel. See also *Figure 2—figure supplement 1*.

The following figure supplement is available for figure 2:

**Figure supplement 1.** Fixed and polymorphic L1HS-Ta elements mapped by ATLAS-seq.

the L1 loci are present in all tested cell lines and approximately 40% of them are present in only one of the cell lines. The remaining insertions show an intermediate level of polymorphism among the studied cell lines (*Figure 2—figure supplement 1c*). Finally, each pair of distinct diploid normal fibroblast lines (IMR-90, BJ and MRC-5 in our panel) differs at an average of 298 positions with regards to the presence or absence of a specific L1HS-Ta copy, in remarkable agreement with previous estimates (*Figure 2—figure supplement 1d*) (*Ewing and Kazazian, 2010*). Collectively, these data reinforce the notion that L1HS-Ta elements are highly polymorphic and contribute to the diversity of the human genome.

## Identification of a molecular signature of transcribed L1HS-Ta copies in MCF7 cells

Full-length L1 elements contain internal sense and antisense promoters located in their 5'-untranslated region (UTR). To gain insight into the pattern of L1 expression in human somatic cells at single copy resolution, we noted that the weak L1 polyadenylation signal in the 3' UTR allows a fraction of

L1 transcripts to extend into the downstream genomic sequence (*Holmes et al., 1994*). This property enables us to use these 3' downstream transcripts as a measure of L1 transcriptional activity, thereby circumventing the difficulties associated with attempting to unambiguously determine the origin of transcript sequences derived from within the highly-identical L1 elements (*Figure 3a* and *Figure 3—figure supplement 1*). Similarly, the L1 antisense promoter activity generates antisense transcripts extending into the upstream genomic region flanking the L1 element (*Speek, 2001*; *Cruickshanks and Tufarelli, 2009*; *Rangwala et al., 2009*; *Macia et al., 2011*; *Denli et al., 2015*). To identify such transcripts, we performed paired-end (2x150 bp) and stranded poly(A)+ RNA sequencing (RNA-seq, *Supplementary file 2*) from the highly L1-expressing breast cancer cell line MCF7 (*Figure 1*). Then we used sense RNA-seq tags downstream of the L1 elements mapped by ATLAS-seq, or antisense RNA-seq tags upstream of them, as a proxy to monitor the sense and antisense promoter activities of each individual copy, respectively (*Figure 3a*).

This strategy enabled us to clearly identify individual, expressed copies of full-length L1HS-Ta, including both fixed and polymorphic instances, as exemplified by the two loci depicted in *Figure 3b*. The first one is a reference L1HS-Ta element integrated in the *TTC28* gene (22q12.1). Uniquely mapped RNA-seq tags are detected both in the body of the L1 sequence and in the immediate downstream genomic region. However, read mapping is performed against the reference human genome; thus, reads mapping within the L1 body could originate either from this particular reference L1 copy or from a non-reference L1 copy integrated somewhere else in the genome, and are therefore not informative. The second example is a non-reference L1HS-Ta element inserted in the *NEDD4* gene (15q21.3) in opposite orientation. Again, a downstream RNA-seq peak immediately follows the insertion point. Interestingly, in both cases, we do not detect upstream antisense RNA-seq tags at the vicinity of these full-length L1HS-Ta copies, suggesting that sense and antisense L1 promoter activities - or the stability of their respective transcripts - are not necessarily coupled in a chromosomal environment, in agreement with previous results obtained from plasmid borne reporter assays (*Macia et al., 2011*).

Several lines of evidence confirmed that the downstream RNA-seq peaks emanate from these L1 copies and not from other overlapping genic or non-genic transcripts, or from distinct L1 copies carrying a 3' transduction (see below). First, we observed H3K4me3 ChIP-seq peaks immediately upstream and adjacent to these L1 copies, a histone mark reflecting active or poised promoters (*Figure 3a–b*) (*Bernstein et al., 2012*). Although the H3K4me3 signal is expected to be centered on the internal promoter within each L1 copy, a region that is either non-uniquely mappable or not included in the reference genome sequence, the ChIP-seq signal can be readily detected in the flanking genomic sequence. Second, we performed shRNA-mediated knockdown of all L1HS-Ta transcripts, targeting the ORF1 sequence (*Figure 3c*). Two different ORF1 shRNAs greatly reduced the downstream RNA-seq signal when compared to a scrambled shRNA or to unmanipulated cells (*Figure 3d*). Together, these data indicate that downstream RNA-seq tags originate from the same transcriptional unit as the considered L1. Retrotransposition of L1 transcripts which include downstream genomic sequences can result in duplication of these sequences at the new insertion site, a phenomenon termed 3'-transduction (*Holmes et al., 1994*; *Moran et al., 1999*). Thus, it is possible, in principle, that downstream RNA-seq tags mapping to a particular L1HS-Ta copy could in fact emanate instead from a daughter copy with a 3' transduction, located elsewhere in the genome. However, the concomitant presence of an upstream H3K4me3 mark at many transcriptionally active L1 copies renders this situation very unlikely in most cases.

## L1HS-Ta transcription originates from a small number of cell-type-specific loci

To obtain a comprehensive view of the transcriptional activities of individual L1 copies, we applied this integrative approach to all the full-length L1HS-Ta elements identified by ATLAS-seq in MCF7 (*Figure 3e*). Strikingly, only 5 L1HS-Ta copies show relatively high expression, and approximately 15 more copies exhibit low but detectable levels of expression. In addition, only 4 L1HS-Ta loci show evidence of L1 antisense promoter activity, including instances that are distinct from the few copies expressing high levels of L1 sense transcripts, suggesting uncoupling and differential regulation between these two L1 promoter activities. Consistent with the examples described at the *TTC28* and *NEDD4* loci (*Figure 3b*), transcribed L1HS-Ta loci at the genome-wide level have a dual fingerprint with upstream active chromatin marks (H3K4me3 and H3K27ac) and Pol-II and downstream

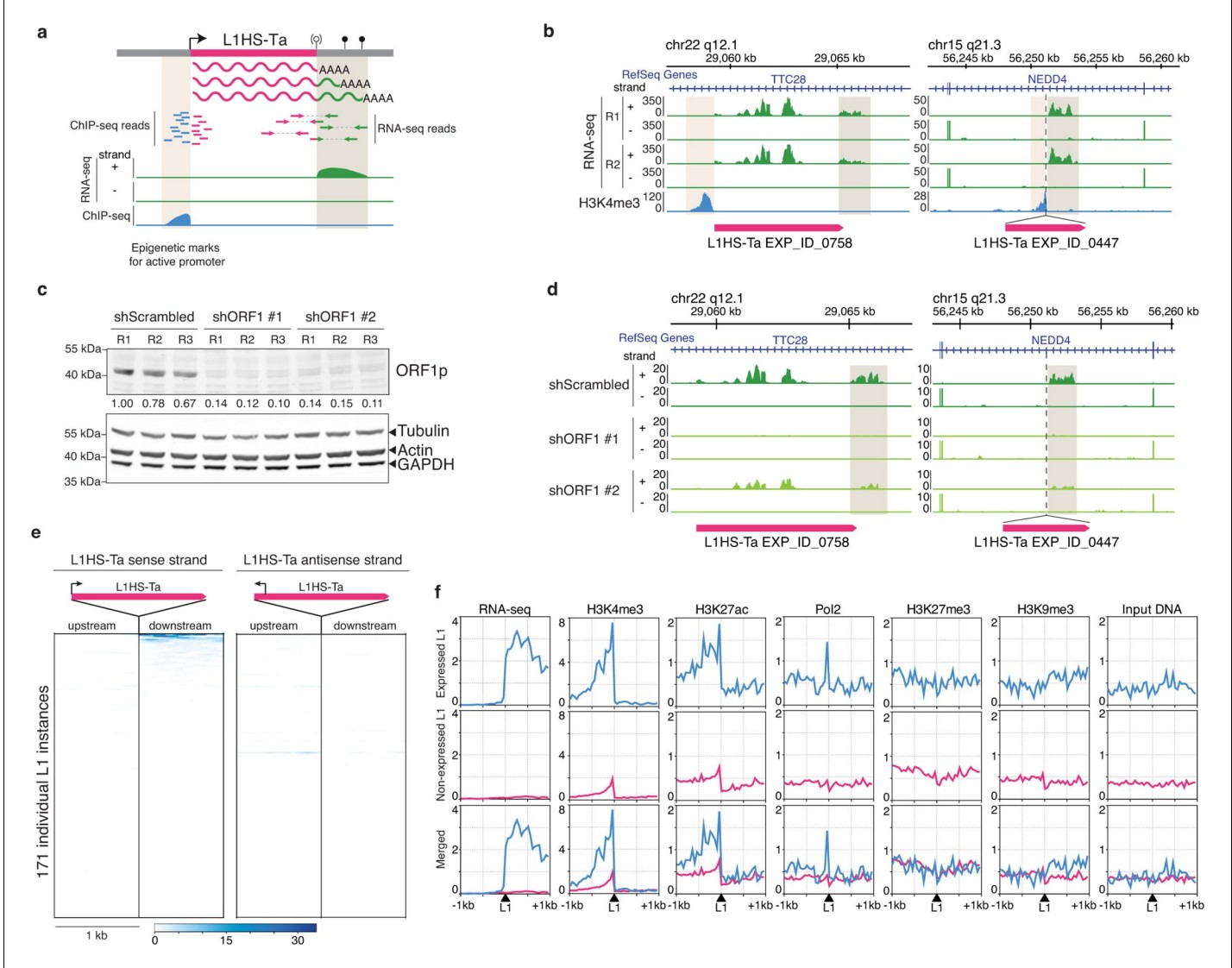

**Figure 3.** Detection of transcriptionally active L1HS-Ta elements at individual copy resolution in MCF7 cells. (a) Theoretical scheme representing the outcome of RNA-seq and ChIP-seq read mapping at polymorphic L1 loci. The informative regions are highlighted in beige. (b) Genome browser views of reference (left, *TTC28* locus) and non-reference (right, *NEDD4* locus) L1 instances integrated with RNA-seq (green) and H3K4me3 ChIP-seq data (blue). R1 and R2, replicate #1 and #2, respectively. (c) shRNA-mediated ORF1p knock-down. Top, immunoblot for ORF1p. Bottom, immunoblot for Actin, Tubulin and GAPDH as loading controls. R1, R2, and R3 are independent knock-down replicates performed in parallel and used subsequently for RNA-seq. Relative ORF1p levels normalized by the loading controls and scrambled shRNA controls are indicated between the two membranes. (d) Modified IGV genome browser views (**Thorvaldsdóttir et al., 2013**) of the *TTC28* (left) and *NEDD4* (right) L1 instances with RNA-seq data upon ORF1p shRNA-mediated knock-down. The informative L1 downstream region is highlighted in beige. Only one biological replicate out of three is shown for the sake of clarity. (e) Heat maps showing RNA-seq read accumulation 1 kb upstream and 1 kb downstream of each L1 copy. The downstream signal on the L1 strand (left heat map) is indicative of L1 sense promoter activity, while the upstream signal on the L1 antisense strand (right heat map) reflects L1 antisense promoter activity. L1 instances (rows) are sorted by decreasing L1 level of expression on the sense strand and the order is identical for the antisense strand. (f) Chromatin and transcription status around expressed (blue, FPKM of downstream RNA-seq tag>0.05) and non-expressed L1HS-Ta instances (pink). The indicated ChIP-seq and RNA-seq signals for each class of L1HS-Ta copies were aggregated and plotted centered around the position of the L1 insertion site. Note that the internal L1 region, when available (reference L1), is not included, but only its flanks. See also *Figure 3— figure supplements 1* and *2*.

The following figure supplements are available for figure 3:

**Figure supplement 1.** L1HS-Ta loci belongs to low mappability genomic regions.

**Figure supplement 2.** Impact of shRNA-mediated ORF1 knockdown on RNA levels for each L1HS-Ta genomic instance.

RNA-seq signal (*Figure 3f*); the latter being reduced upon shRNA-mediated L1 knockdown (*Supplementary file 2* and *Figure 3—figure supplement 2a*). The expression pattern of individual L1HS-Ta loci in standard growth conditions is highly reproducible, as revealed by the clustering of independent RNA-seq experiments obtained from the same cell line grown in two independent laboratory environments (*Figure 4a*, MCF7_Cristofari and MCF7_ENCODE samples, Pearson correlation r=0.830 p<0.0001), highlighting the non-random nature of this process and the robustness of our overall approach.

To expand these observations to a range of cell-types we applied the same RNA-seq-based analysis strategy to the complete panel of cell lines (except MRC-5 and HEK-293 cells for which stranded poly(A)+ RNA-seq data were not available in public databases, *Figure 4—figure supplement 1*). The identity of the individual L1HS-Ta loci which are expressed, as well as their levels of expression, varies considerably between cell types (*Figure 4a* and *Supplementary file 5*). Strikingly, many L1HS-Ta elements which are present at the same genomic location in distinct cell types are differentially expressed, indicating that L1HS-Ta re-activation in transformed cells may result from cell-type- and copy-specific regulation.

As for MCF7, L1HS-Ta expression profiles between biological replicates of all cell lines are similar; and global shRNA-mediated ORF1 knockdown in 2102Ep, followed by RNA-seq, again confirmed the L1-derived origin of downstream transcripts in this cell line (*Supplementary file 2* and *Figure 3—figure supplement 2b*). The total levels of RNA-seq tags 1 kb downstream of L1HS-Ta for a given cell line are highly correlated with the number of internal L1 reads containing the ACA diagnostic nucleotides (Spearman r=0.8169, p<0.0001), reinforcing the idea that RNA-seq tags downstream of L1 can be used as a reliable proxy for L1 sense transcription. RT-PCR validation using primers anchored in the 3' UTR of L1 and in the downstream flanking genomic sequence confirmed the transcriptional state deduced from RNA-seq, for a selection of expressed and non-expressed L1HS-Ta elements (*Figure 4—figure supplement 2*). Interestingly, for most L1HS-Ta-expressing cells, only 5 to 15 individual copies contribute to the bulk of L1HS-Ta RNA, defined as the number of L1 copy contributing to half of the total FPKM count (*Figure 4b*). As compared to MCF7 cells, the embryonal carcinoma cells 2102Ep accumulate comparable global levels of L1HS-Ta transcripts, but seem to have a higher number of permissive L1HS-Ta loci, each contributing to a smaller proportion of the total, although the number of active instances still represents a small fraction (<10%) of all L1HS-Ta copies in these cells.

## Relationship of expressed L1HS-Ta copies with genes and copy-number variants

To test whether the expression of L1HS-Ta copies could be influenced by their genic environment, we first compared the proportion of expressed L1 insertions in genes as compared to non-expressed copies. Approximately 1/3 of all full-length L1HS-Ta copies were inserted in genes (*Figure 4c*). Although the proportion of expressed L1s in genes was slightly higher than that of non-expressed copies, the difference was not significant. Then, we focused on the genic cohort of full-length L1HS-Ta. For the latter, we asked whether expressed *vs.* non-expressed elements were differentially oriented relative to the overlapping genes. Consistent with previous observations (*Szak et al., 2002*), genic L1 copies are more often found in the antisense orientation (*Figure 4—figure supplement 3a*). However, this proportion was not significantly different between expressed and non-expressed copies. Independently of their orientation relative to genes, we found that genes containing expressed L1 are often more expressed than those containing non-expressed L1 (*Figure 4d*), suggesting that highly expressed gene loci might represent a favorable genomic environment for L1 reactivation. Finally, it is conceivable that expressed L1 could be located in larger chromosomal regions having undergone massive amplification. To address this possibility, we looked whether L1 insertions were located in genomic regions showing copy number variations (CNVs). The majority of L1 copies were inserted in normal regions (*Figure 4—figure supplement 3b*), whether they were expressed or not, and expressed L1 copies were not significantly enriched in amplified regions.

## Many transcribed L1HS-Ta are retrotransposition-competent

Finally, we determined whether the top expressed L1HS-Ta elements identified as expressed in the panel of cell lines have the ability to achieve complete retrotransposition cycles and to generate new

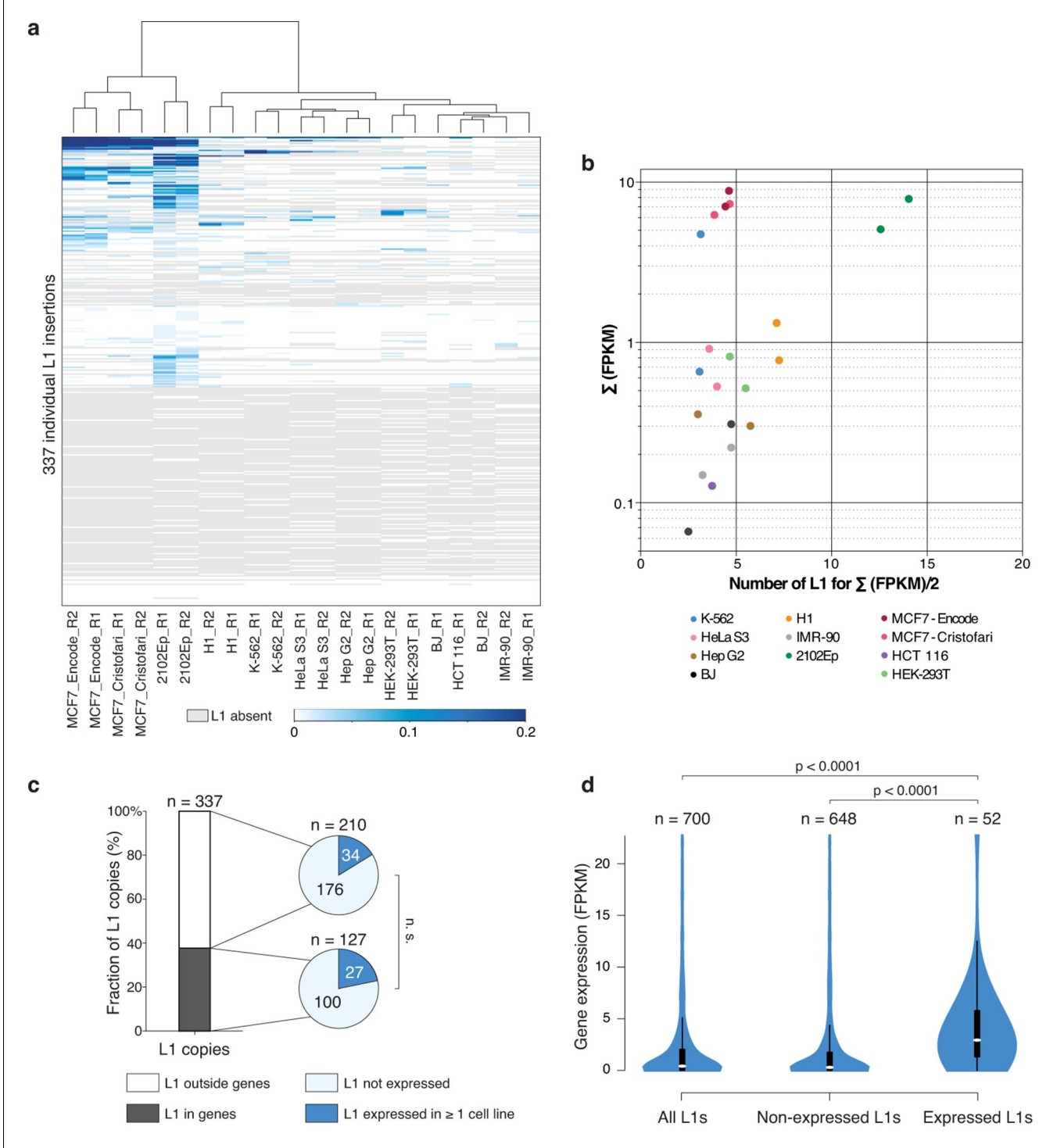

**Figure 4.** Locus- and cell-type-specific reactivation of individual L1HS-Ta copies in normal and transformed cells. (**a**) Heat map displaying expression levels of each L1 instance in each of the analyzed cell lines. Expression level is defined as the number of RNA-seq fragments mapped in a 1 kb-window downstream of a particular L1 copy and on the same strand, normalized by the total amount of mapped fragment (FPKM). Grey, absent polymorphic L1 copy. Most cell lines have at least two RNA-seq replicates (R1 and R2), which cluster based on their L1 expression profiles, showing their cell-line specificity. (**b**) The bulk of L1HS-Ta transcripts is produced by a limited number of loci. Scatter plot showing the number of L1 copies contributing to half of the total pool of L1HS-Ta transcripts. The y-axis represents the total L1 downstream tag FPKM count for each cell line. The x-axis represents the number of L1 loci contributing to half of this total FPKM. (**c**) Distribution of expressed and non-expressed L1 insertions in genic and non-genic regions. Bar chart indicating the fraction of L1 copies in genic (dark grey) and non-genic (white) regions with associated pie charts indicating the proportion of

*Figure 4 continued on next page*

*Figure 4 continued*

non-expressed L1 (light blue) and L1 expressed in at least 1 cell line of the panel (dark blue). The distribution of expressed L1 insertions is not statistically different between genic and non-genic regions (p=0.117, binomial test). (d) Expression levels of genes associated with non-expressed or expressed L1 copies. Values of gene expression are considered independently for each cell line of the panel, and distributed whether the L1 insertion is expressed (>0.05 FPKM) or not (≤0.05 FPKM) in each particular cell line. White oval shows the median; black box lower and upper limits indicate the 25th and 75th percentiles, respectively; whiskers extend to 1.5 times the interquartile range; violin shape represents density estimates of data and extend to extreme values (out of scale range). Genes containing expressed L1s are more expressed than genes containing non-expressed L1s (p<0.001, Kolmogorov-Smirnov test). See also *Figure 4—figure supplements 1*, *2* and *3*.

The following figure supplements are available for figure 4:

**Figure supplement 1.** Heat maps for L1HS-Ta loci individual expression in various cell lines.

**Figure supplement 2.** RT-PCR validation of individual L1 expression across several cell lines.

**Figure supplement 3.** Relationship of expressed L1HS-Ta copies with genes and ploidy.

copies. To answer this question, we combined complementary strategies. First, we collated published data of retrotransposition assays in cultured cells obtained for different L1 instances (*Brouha et al., 2003*; *Beck et al., 2010*), and combined them with our own experimental results obtained with an additional newly-identified L1 copy following the same protocol (*Figure 5b*). Second, we compared our set of highly-expressed L1HS-Ta copies with those which have been identified in earlier studies as mobilization-competent based on the detection of daughter copies with matching 3'-transduced sequences (*Tubio et al., 2014*). Third, and most directly, we specifically searched for evidence of 3' transduction in our 3' ATLAS-seq data using split reads partly mapping downstream of two distinct L1HS-Ta copies (see Materials and methods and *Figure 5c*). We found that 5 out of the 20 most highly-expressed L1HS-Ta copies across all cell lines fulfill at least two of these criteria strongly supporting their ability to retrotranspose and 6 additional ones could be identified as progenitors of other, daughter copies. The remaining nine elements could also be retrotransposition-competent, but have not been tested in cultured assays, nor could daughter L1 copies be unambiguously identified. Thus, in total, at least 11 out of the 20 most highly-expressed L1HS-Ta copies across all cell lines are retrotransposition-competent (*Figure 5a* and *Supplementary file 5*).

Some L1 elements with high retrotransposition activity ('hot' L1) belong to well-defined lineages with distinctive 3' transductions. To evaluate the proportion of the highly expressed elements which belong to such lineages, we screened 3' ATLAS-seq reads supporting each L1 insertion for sequence tags characteristic of the three most characterized lineages (AC002980, LRE3 and RP, see Methods) (*Schwahn et al., 1998*; *Brouha et al., 2002*; *Myers et al., 2002*; *Beck et al., 2010*; *Macfarlane et al., 2013*). We found only 1 insertion (EXP_ID_0447, in the *NEDD4* gene) among the 20 most highly expressed L1HS-Ta copies as deriving from the L1$_{RP}$ transduction family (*Supplementary file 5*). Nine additional copies were also part of one or another lineage, but were not expressed – or only moderately – in any of the cell lines analyzed. Thus, these findings suggest that the observed high level of expression of a small cohort of L1 insertions is not an intrinsic feature of any previously identified lineage.

## Discussion

Altogether our observations support a model (*Figure 6*) where: (i) L1HS-Ta transcription is predominantly inactive in somatic human cells, including transformed cell lines; however (ii) a small number of L1HS-Ta copies can escape silencing, allowing their expression and transcript accumulation; (iii) the locus in which a particular L1HS copy integrates has a major influence on its ability to be subsequently reactivated; (iv) L1 instances at distinct genomic loci are subject to cell-type dependent activation (potentially dependent on environmental or physiological signals). This model is consistent with previous observations made on few L1 instances suggesting that the transcriptional activity of the L1 promoter might be influenced by its immediate upstream genomic sequence (*Lavie et al., 2004*).

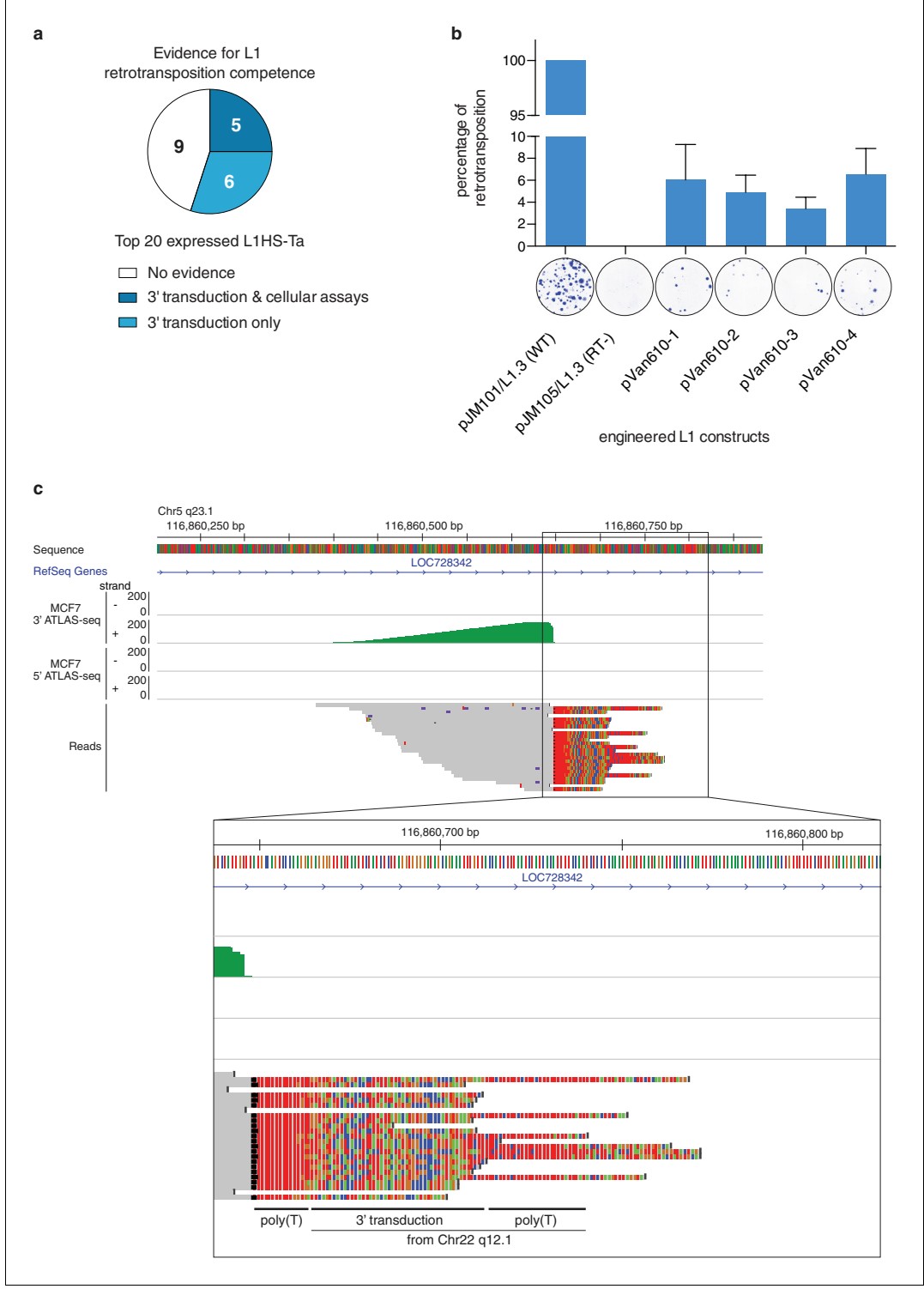

**Figure 5.** Evidence of retrotransposition capability for selected L1HS-Ta copies. (**a**) Evidence of retrotransposition competence for the top 20 most expressed L1 copies across all cell lines analyzed. Cellular assays refer to retrotransposition cellular assays of plasmid-borne L1 instances, whose expression is driven by either the native L1 5' UTR alone (***Brouha et al., 2003***) or supplemented by a strong CMV promoter ([***Beck et al., 2010***] and ***Figure 5b***). These assays measure L1 intrinsic biochemical activity, independently of their actual expression in their genomic context. Three-prime transduction refers to the existence of progeny copies containing a 3' transduction, which can be traced back to the original locus and reflect a retrotransposition event. (**b**) Retrotransposition assay in cultured cells for MCF7 L1 copy EXP_ID_0447 (*NEDD4* locus). A full length transcribed L1HS-Ta copy present in

*Figure 5 continued*

the genome of MCF7 cells was subcloned by PCR in an expression vector containing a reporter gene to measure retrotransposition activity and generated four independent clones (pVan610-1 to -4). In transfected HeLa cells, *de novo* retrotransposition events of engineered L1 copies lead to the introduction of a functional genomic copy of the neomycin phosphotransferase gene, which expression confers resistance to G418. Resistant foci were stained and counted to monitor retrotransposition activity compared to the positive (pJM101/L1.3, wild type L1HS-Ta) and negative (pJM105/L1.3, mutant L1HS-Ta) control conditions. The value of G418 resistant colonies obtained with the positive control was set to 100%. A picture of a representative well with stained colonies is displayed for illustrative purposes under each bar of the graph. The average value of three biological replicates is displayed with error bars corresponding to the standard deviation among the three biological replicates. (c) Detection of 3' transductions in ATLAS-seq data. This in silico screen identifies L1HS-Ta copy (progeny element) with ATLAS-seq clusters containing reads with non-aligning subsequences (soft-clipped), which uniquely map downstream and adjacent to another full length L1HS locus (progenitor element). The panel shows a genome browser view of such a 3' transduction, originating from a full length L1HS-Ta in the *TTC28* gene (22q12.1). The soft-clipped region of the reads is shown in color (base code: T, red; A, green; C, blue; G; orange). As expected, the transduced region is flanked by 2 poly(A) tails (poly(T) here since it is located on the reverse genomic strand).

L1 retrotransposon expression has been proposed both as a potential biomarker of cancer prognosis and as the starting point for L1-mediated genome instability in tumors (*Piskareva et al., 2011*; *Rodić and Burns, 2013*; *Rodić et al., 2014*). Hence, understanding the means by which L1s can escape regulation in particular cancers is vital in order to improve their rational use as biomarkers or to predict their possible effects on disease progression. Strikingly, our results indicate that – in the cancer cell-lines studied – the general cellular regulation of most L1 instances is unimpaired, even in cells exhibiting high L1 activity. Thus, it appears that shut-off of global L1-regulatory pathways is not a prerequisite for L1 activation in cancer. Instead, we find that only a very limited set of L1 copies, at specific genomic loci, become activated in cancer cells. We note that the presence or absence of polymorphic L1s at these permissive loci, and their degree of transcriptional activation, may represent risk factors for particular cancer types, and more specific biomarkers than global L1 expression or methylation status.

Furthermore, we provide here a unique resource consisting of near-complete maps of L1HS-Ta elements present in widely used normal and transformed model cell lines, for which the availability of genomic datasets is regularly increasing (including several tier-1 & tier-2 cell-lines of the ENCODE project). These maps will be of broad utility in the future to address the impact and regulation of

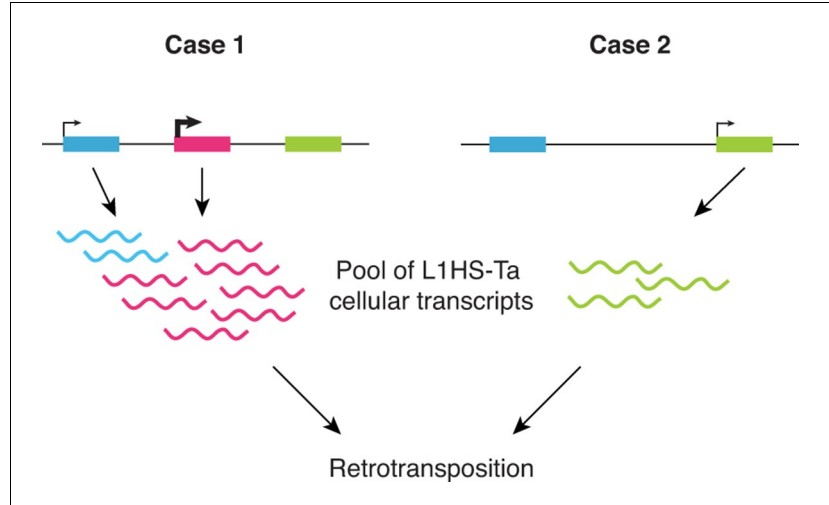

**Figure 6.** Schematic model showing the highly locus-specific and variable expression of L1HS-Ta elements among different somatic cell types and individuals. The colored boxes correspond to L1HS-Ta copies, some being polymorphic (pink). The model is developed in the main text.

transposable elements in the human genome. Indeed, isolated RNA-seq and ChIP-seq signals resulting from the presence of non-reference L1 copies can only be correctly interpreted if such maps are available. Thus they can act as a platform to fully interpret and profit from the many expanding public datasets generated using the same cell lines.

Recently, whole genome sequencing of human tumors has revealed recurrent retrotransposition events stemming from a handful of source elements (*Tubio et al., 2014*), consistent with the notion that only a fraction of all full-length L1 elements is actually capable of retrotransposition (*Brouha et al., 2003*; *Beck et al., 2010*). Here, we have identified highly heterogeneous expression of individual L1HS-Ta copies, implicating L1 transcriptional activation as a key regulatory process which limits the mutagenic potential of L1 elements independently of their respective intrinsic biochemical activity. Therefore, we extend the concept of retrotransposition-competent L1 copies, previously described as 'hot L1s' (*Brouha et al., 2003*; *Beck et al., 2010*), to the transcriptional regulation of each individual locus and cell type. We conclude that L1-mediated mutagenesis results from the reactivation of a small subset of permissive loci, only a fraction of which contains retrotransposition-competent elements, combined with a favorable cellular environment (*i.e.* diminished restriction factor and/or increased cofactor activities [*Pizarro and Cristofari, 2016*]). Several of these regulated loci are polymorphic with regards to the presence or absence of an L1 element among the human population, highlighting the role of genetic determinants in the global L1 mutagenic potential in a given individual. Overall, our data suggest that activation of L1 transcription in somatic cells is governed by individual-, locus-, and cell-type-specific determinants and provide a framework to study how distinct L1HS-Ta copies may be regulated by environmental, physiological and pathological triggers. Future work will determine the factors and cellular signaling pathways that contribute to the transcriptional reactivation of the different L1HS-Ta copies in somatic cells.

## Materials and methods

### Cell culture

The cell lines used in this study are summarized in *Supplementary file 1*. Cells were grown in the medium indicated in *Supplementary file 1*, containing 4.5 g/L D-Glucose, 110 mg/L Sodium Pyruvate, and supplemented with 10% FBS, 100 U/mL penicillin and 100 µg/mL streptomycin. Growth medium was also supplemented with 862 mg/mL L-Alanyl-L-Glutamine (Glutamax), or 2mM Glutamine (HCT 116 and K-562). The HEK-293 cell line is a clonal derivative of primary human embryonic kidney cells transformed with Adenovirus 5 (Ad5) DNA (*Graham et al., 1977*). The HEK-293T cell line (also known as 293T or 293/tsA1609neo) is a clonal derivative of HEK-293 cells stably transfected with a temperature sensitive SV40 Large T antigen allele and the neomycin resistance gene (*DuBridge et al., 1987*). All above-mentioned cell cultures tested negative for mycoplasma infection using the MycoAlert Mycoplasma Detection Kit (Lonza, Basel, Switzerland). Cell line authenticity was verified by multiplex STR analysis (PowerPlex 21 PCR system [Promega, Madison, WI], assays performed by Eurofins Genomics [Ebersberg, Germany] as a service provider) and comparison with the DSMZ database (https://www.dsmz.de/services/services-human-and-animal-cell-lines/online-str-analysis.html) or with previously published profiles for H1 and 2102Ep cells (*Josephson et al., 2007*; *Mallon et al., 2014*).

### Oligonucleotides

Oligonucleotides were synthesized by Sigma-Aldrich (St Louis, MO), Eurogentec (Liège, Belgium) or Integrated DNA Technologies (IDT, Coralville, IA). Those used for ATLAS-seq, shRNA constructs, PCR or RT-PCR assays are described in *Supplementary file 2* and those used to validate ATLAS-seq results are described in *Supplementary file 4*.

### Protein extracts and RNP preparation

For whole cell lysates, $3 \times 10^6$ cells were pelleted and lysed in 150 µL RIPA buffer (10 mM Tris-HCl pH 7.5, 1 mM EDTA, 150 mM NaCl, 0.5% NP-40, 0.5% sodium deoxycholate, 0.1% SDS, supplemented with complete Mini, EDTA-free Protease Inhibitor Cocktail tablets [Roche, Basel, Switzerland]) by 30 cycles of sonication (30 s on/30 s off) at 4°C with a Bioruptor sonicator (Diagenode, Liège, Belgium). Protein concentration was determined by the BCA assay (Interchim, Montluçon, France). L1

ribonucleoprotein particles (RNP) were enriched by ultracentrifugation on sucrose cushion as described previously (*Kulpa and Moran, 2006*; *Monot et al., 2013*; *Viollet et al., 2016*). Briefly, $10^7$ cells were collected and washed with 1X phosphate-buffered saline (PBS). Cell pellets were lysed with 1 mL of LEAP buffer (5 mM Tris-HCl pH 7.5, 1.5 mM KCl, 2.5 mM MgCl$_2$, 1% sodium deoxycholate, 1% Triton X-100, supplemented with complete Mini, EDTA-free Protease Inhibitor Cocktail tablets [Roche]) overnight at 4°C on an end-over-end wheel. Cell lysates were loaded on top of a two-layered sucrose cushion (8.5% and 17% sucrose solution diluted in 80 mM NaCl, 5 mM MgCl$_2$, 20 mM Tris pH 7.5 and 1 mM DTT supplemented with complete, Mini, EDTA-free Protease Inhibitor Cocktail tablets). Samples were ultracentrifuged at 39,000 rpm in a Beckman SW 41 Ti rotor for 2 hr at 4°C and resuspended overnight in 50 µL of MilliQ water. Protein concentration was measured by fluorometry using the Qubit protein assay kit (Life Technologies, Carlsbad, CA).

## Immunoblots

Unless otherwise stated, 30 µg of whole cell lysates or 10 µg of RNP were resolved on a 4–12% gradient NuPAGE Bis-Tris gel (Life Technologies) and transferred with the semi-dry XCell II Blot module onto PVDF FL membrane (Millipore, Billerica, MA). Membranes were incubated for 1 hr at room temperature in blocking solution (Phosphate-buffered saline with 0.1% Tween 20 (PBS-T), containing 5% (w/v) fat-free milk), and then overnight at 4°C with a primary antibody diluted in blocking solution. After 5 washes in PBS-T, the membranes were incubated for 1 hr at room temperature with the appropriate secondary antibody coupled to infrared fluorochromes diluted in Odyssey blocking buffer (LI-COR Biosciences, Lincoln, NE) and washed again 5 times in PBS-T and once in PBS for 5 min. The signal was detected and quantified with a dual-channel Odyssey infrared imaging system (LI-COR Biosciences). When needed, membranes were stripped for 1 hr in LI-COR stripping buffer, washed 3 times in MilliQ water and were reprobed for loading control following the protocol described above. Primary antibodies were directed against hORF1p (rabbit polyclonal antibody, serum SE-6798, 1:5,000) (*Monot et al., 2013*), S6 Ribosomal Protein (RPS6, rabbit monoclonal antibody, clone 5G10, Cell Signaling Technology [Danvers, MA], 1:2500), β-Tubulin (TUBB, mouse monoclonal antibody, clone BT7R, Pierce Biotechnology [Waltham, MA], 1:5000), β-Actin (ACTB, mouse monoclonal antibody, clone 8H10D10, Cell Signaling Technology, 1:5000), and Glyceraldehyde-3-phosphate dehydrogenase (GAPDH, mouse monoclonal antibody, clone GA1R, Pierce Biotechnology, 1:5000). As secondary antibodies, we used IRDye 800CW Goat anti-Rabbit IgG, IRDye 800CW Goat anti-Mouse IgG, IRDye 680RD Goat anti-Rabbit IgG, or IRDye 680RD Goat anti-Mouse IgG (all from LI-COR Biosciences, 1:10,000).

## Total RNA and genomic DNA preparations

Whole cell total RNA was purified by a double TRI Reagent extraction following the manufacturer's instructions (Molecular Research Center, Cincinnati, OH) and solubilized in 50 µL of milli-Q water. Subsequently, 20 µg of total RNA was treated with 2U of TURBO DNAse (Life technologies) for 20 min at 37°C followed by a 5 min incubation step at room temperature with DNase Inactivation Reagent. After centrifugation at 10,000 x g for 1.5 min, the supernatants containing the RNA samples were transferred into a new tube. RNA was quantified and quality-controlled by UV-spectroscopy (NanoDrop 2000) and microfluidic electrophoresis (Agilent 2100 Bioanalyzer), respectively. For RNA-seq samples an additional column clean up was performed (RNeasy mini extraction kit, Qiagen, Hilden, Germany). Genomic DNA was extracted and prepared using QiaAmp DNA Blood mini kit (Qiagen).

## ATLAS-seq library preparation and sequencing

The primers used in the ATLAS protocol (*Badge et al., 2003*) allow the specific amplification of the 3' flank of the L1HS-Ta subfamily (3'-ATLAS) and of the 5' flank of the L1HS-Ta1d subfamily (5'-ATLAS). While representing less than 0.1% of human L1s, the L1HS-Ta and L1HS-Ta1d subfamilies are the youngest and most active elements (*Boissinot et al., 2000*) and are responsible for all but one described disease-producing insertions (*Chen et al., 2005*). ATLAS-seq differs from the original ATLAS protocol in three main aspects: (i) Restriction enzyme digestion of the genomic DNA was replaced by mechanical fragmentation to reduce sequence bias and thus to increase the mappable part of the genome. DNA is end-repaired and A-tailed to permit ligation of the adapter to

fragmented genomic DNA, before the suppression PCR step; (ii) Adapters have been modified to be compatible with A-tailed genomic DNA fragments and to include Ion Torrent primer sequences; (iii) Cloning and Sanger sequencing were replaced by Ion Torrent semiconductor sequencing. The full ATLAS-seq protocol and analysis scheme are detailed below.

## Mechanical fragmentation, end-repair and A-tailing

One microgram of genomic DNA was sonicated for 6 cycles (6 s on/90 s off) at 4°C with a Bioruptor sonicator (Diagenode), generating average fragments of 1 kb. DNA ends were repaired using the End-It DNA End-Repair Kit (Epicentre, Madison, WI). A-tailing of the repaired blunt ends was performed with Klenow Fragment (3'-to-5' exo-, New Englands Biolabs, Ipswich, MA) following the manufacturer's protocol. Between each step, DNA was purified with Agencourt AMPure XP beads (Beckman Coulter, Brea, CA) using a 1.8:1 ratio of beads to DNA solution (v/v).

## Linker ligation

The LOU365 and LOU366 linker oligonucleotides were mixed in 50 µL of 1x T4 DNA Ligase buffer (50 mM Tris-HCl, 10 mM MgCl$_2$, 1 mM ATP, 10 mM Dithiothreitol, pH 7.5, New England Biolabs) at a final concentration of 80 µM each and annealed by heating at 65°C for 15 min followed by slow cooling down to room temperature. DNA fragments (1 µg) were ligated with a 40-fold molar excess of annealed linkers overnight at 16°C in 50 µL of 1x T4 DNA Ligase buffer supplemented with 400 U of T4 DNA Ligase (New England Biolabs). Excess linkers were removed by purification with Agencourt AMPure XP beads.

## Suppression PCR

To reduce PCR stochasticity, the ligated genomic DNA of each sample was amplified in 8 independent parallel reactions of 40 µL each, containing 9 ng of ligated genomic DNA, 0.2 µM of primers, 0.2 µM dNTPs, 1.5 mM MgCl$_2$, 1X PCR buffer, and 1 U of Platinum Taq DNA Polymerase (Invitrogen, Carlsbad, CA). Amplification was performed under the following cycling conditions: 1 cycle at 95°C for 4 min; followed by 18 cycles at 95°C for 30 s, 64°C for 30 s, and 72°C for 1 min; and a final extension step at 72°C for 7 min. Primers are described in *Supplementary file 2*.

## Size selection

PCR products corresponding to the same sample were pooled and subjected to a double size selection using two consecutive Agencourt AMPure XP bead purifications using beads to DNA ratios of 0.6:1 and 0.7:1, respectively. The supernatant of the first bead purification contains DNA fragments smaller than 550 bp and is applied to the second step, where fragments larger than 450 bp are retained on the beads and subsequently eluted in 20 µL. Each library is quality-controlled by Bioanalyzer 2100 (DNA high sensitivity kit, Agilent Technologies, Santa Clara, CA) and quantified using a qPCR based assay (library quantification kit for Ion Torrent, Kappa Biosystems, Wilmington, MA).

## Ion Torrent PGM sequencing

For sequencing, libraries were pooled two-by-two in equimolar amounts (final concentration of 20 pM). Emulsion PCR and enrichment for positive Ion Sphere Particles (ISPs) was performed on the Ion OneTouch 2 and ES enrichment modules, respectively, using the Ion PGM Template OT2 400 Kit (Life Technologies), and sequenced on the Ion Torrent PGM, using the Ion PGM Sequencing 400 Kit and Ion 318 v2 Chips (Life Technologies), according to the manufacturer's protocols. ATLAS-seq sequencing statistics are summarized in *Supplementary file 2*.

## ATLAS-seq bioinformatic analysis

The analysis was performed using dedicated scripts starting from raw reads in fastq format. The different steps are detailed below.

## Read preprocessing

In brief, the reads were trimmed to remove barcodes, low-quality sequence, ATLAS-seq primers and linkers, L1 sequence and poly(A) tail, and to keep only the putative genomic flanking sequences. Sequences shorter than 25 nt after trimming were excluded.

More specifically, we first demultiplexed fastq files according to the sample-specific barcode and using the FASTX-Toolkit (http://hannonlab.cshl.edu/fastx_toolkit/). Then we treated differently reads produced in 5'- and 3'-ATLAS-seq experiments.

For 3'-ATLAS reads, we first trimmed barcode sequence at the 5' end, allowing only perfect matches using cutadapt (-e 0 -m 25 options) (*Martin, 2011*). Reads without detectable barcode or shorter than 25 bp after trimming were discarded. Second, we trimmed the suppression PCR linker from the 5' end, allowing 5 errors maximum (-e 0.12 -m 25 options). Reads without detectable linker or shorter than 25 bp after trimming were discarded. Third, we removed low quality sequence at the read 3' end (Q<12) and iteratively searched and trimmed at the read 3' end: (i) the RB3PA1 primer, (ii) the L1 3' end sequence, and (iii) a polyA sequence, using cutadapt (-e 0.12 -m 25 -q 12 -O 8 options). For each read, we recorded whether it was trimmed and which feature was trimmed. However, reads without detectable feature were not discarded since they most often corresponded to reads that did not reach the L1 junction.

For 5'-ATLAS-seq reads, since sequencing was initiated from the internal part of L1, we performed an initial size selection to remove reads that did not reach the junction (<133 bp). Then, we iteratively searched and trimmed at the 5' end: (i) the RB5PA2 primer, and (ii) the L1 5' end sequence, followed by the ATLAS linker sequence at the 3' end, using cutadapt (-e 0.12 -m 25 -q 12 options). Reads missing one of these features were discarded. Finally, we reverse complemented read sequence using the FASTX-Toolkit. The later step ensured that both 5'- and 3'-ATLAS reads started from the linker position, information necessary to remove PCR duplicates (see below). Hence, at the end of read preprocessing, all reads were oriented in a convergent manner toward the L1 element (from flanking sequence to L1 body).

## Reads mapping and clustering

Trimmed reads were mapped to the hg19 human reference genome using the Burrows-Wheeler Aligner (BWA) program with the 'mem' algorithm (*Li and Durbin, 2010*) (-t 4 -M options). Mapped reads were filtered to remove secondary alignment and ambiguously mapped reads (MAPQ≤20) using SAMtools (*Li et al., 2009*) (samtools view with -q 20 -F 260 options). Then, PCR duplicate reads were removed with Picard tools (http://broadinstitute.github.io/picard, MarkDuplicates function), keeping only the longest read as a representative. Reads were considered redundant if they started from the same linker position, which corresponds to the initial genomic DNA break during library preparation. However, it should be underlined that this procedure tends to underestimate the number of independent reads, particularly for clusters with an abundant number of reads. We next merged reads with the same orientation which overlapped or were separated by less than 100 bp into clusters using BEDtools (*Quinlan and Hall, 2010*) (bedtools merge with -s -d 100 options). In a given cluster, the vast majority of the reads ended or started at the same position (depending on the orientation of the insertion), which corresponded to the actual insertion site. The position of the putative L1 insertion was defined:

- for insertions in the sense orientation: as the most upstream 5' coordinate for 3' ATLAS clusters and the most downstream 3' coordinate for 5' ATLAS clusters.
- for insertions in the antisense orientation: as the most downstream 3' coordinate for 3' ATLAS clusters and the most upstream 5' coordinate for 5' ATLAS clusters.

At the end of the clustering step, each putative insertion was characterized by a series of features: the genomic coordinates of its 3' end, its orientation, a unique cluster ID, the number of non-redundant reads supporting the insertion, the number of supporting non-redundant reads for which the RB3PA1, L1 and/or polyA sequences were trimmed, the total number of reads supporting the insertion. We also calculated normalized versions of the counts (RPM, read per million of mapped reads for redundant reads; TPM, tags per million of mapped reads for non-redundant reads only).

## Annotating putative insertions with 5'ATLAS clusters, with known L1 insertions and relation with genes

We annotated each putative insertion with a number of features using BEDtools 'closest' function, with a feature-specific window. To identify putative full-length elements, we matched 5'-ATLAS cluster to each putative insertion site in a 7 kb window upstream of the 3' integration

site and in the proper orientation. As expected, the distribution of the distances between the 5′ and 3′ L1 ends was bimodal: either close to 6 kb (L1 element present in the reference genome sequence) or close to 0 (L1 element absent in the reference genome sequence). The position of each putative L1 insertion was also compared to potential spurious RB3PA1 genomic binding sites, to reference L1HS and L1 (UCSC RepeatMasker track, Smit AFA, Hubley R, Green P. *RepeatMasker Open-3.0*. http://www.repeatmasker.org. 1996–2010 and [*Fujita et al., 2011*]), to fixed L1 elements (*Myers et al., 2002*; *Stewart et al., 2011*), to known polymorphic L1HS (euL1db, [*Mir et al., 2015*]), or to Refseq gene positions (*Fujita et al., 2011*). A putative insertion was considered as already known if it fell within the same 200 bp window as another previously described insertion (*Mir et al., 2015*).

## Data filtering

Based on early PCR validations, we considered as true positive a putative insertion supported by at least three 3′-ATLAS-seq non-redundant reads, including at least one ending within a polyA-tract and/or an L1 sequence; or by at least one 5′- and one 3′- ATLAS non-redundant reads.

To obtain high-confidence L1 insertion maps, we implemented a number of additional filters. First, we found that spurious RB3PA1 (partial) match in a non-L1 sequence is a significant source of false positive. These events were characterized by: (i) the presence of an RB3PA1 partial sequence outside of a reference L1 element; and (ii) 3′ ATLAS-seq reads containing the RB3PA1 sequence immediately preceded by the flanking genomic sequence (without L1 or polyA sequence). To filter out these events, we established a map of RB3PA1 partial matches in the human reference genome. We iteratively aligned to the human genome RB3PA1 derivatives, deleted one base after the other from their 5′ end, down to 8 nucleotides, keeping only perfect matches. We also aligned the full RB3PA1 primer (22 nt in length) allowing up to 2 mismatches. Then for each genomic position, if several forms of the primer could match, we kept the longest one. Mapping was achieved with BWA using the 'samse' algorithm (*Li and Durbin, 2010*) and allowing up to $10^9$ possible occurrences (-n 1000000000 option). Putative L1 insertions are considered as false positive - and filtered out - if they combine the following criteria: (i) they are not reference insertions, (ii) they contain a potential partial RB3PA1 primer match and, (iii) their supporting reads contain genomic flanking sequence immediately followed by the RB3PA1 primer sequence (without L1 or polyA sequence). Finally, we observed that L1HS-PreTa copies can sometimes be amplified, but the corresponding clusters are generally supported by 10 times less reads than reference L1HS-Ta. Insertions corresponding to these older elements were also filtered out.

## In silico estimate of the discovery rate

Because reference L1HS-Ta elements are not present in all individuals, they cannot be used to evaluate the overall discovery rate of the ATLAS-seq procedure. Instead, we constructed a reference set of 123 L1HS-Ta copies fixed in the human population. It contains the L1HS-Ta elements previously described as fixed in an early analysis of the human genome (*Myers et al., 2002*), from which we removed a subset of elements, which have been later shown as polymorphic in the frame of the 1000 Genome Project (annotated as 'deleted' as compared to the reference genome, [*Stewart et al., 2011*]), as well as Y chromosome insertions. Irrespective of the cell line and of the in silico filters described in the previous paragraph, we consistently found ≥98% of the fixed insertions (120 to 122 depending on the cell line). We cannot exclude that some of the missing insertions result from overlapping large structural variants, such as the loss of large chromosomal regions, particularly in transformed cell lines.

## PCR validation and estimation of the true positive rate

To evaluate the rate of true positive insertion calls in the overall ATLAS-seq procedure, we selected 72 putative non-reference insertions found in HEK-293T cells with a broad range of supporting read numbers (from 2 to 326, *Supplementary file 4*). This includes 50 previously described polymorphic insertions and 22 totally unknown. We batch designed PCR primers in the 5′ and 3′ flanking sequence spanning a 200 bp region starting 50 nucleotides from the putative insertion site with PrimerQuest online tool (http://eu.idtdna.com/primerquest/home/index) and we verified that each pair generates only one product using the UCSC In-Silico PCR tool (*Kent et al., 2002*). The primer

sequences, based on the hg19 human reference genome sequence, are listed in *Supplementary file 4*. For each tested locus, we performed two PCR reactions with distinct primer pairs. The first one amplified the entire locus using a primer on each side of the putative L1 insertion site. This reaction can result in the amplification of the empty locus and/or the filled locus depending on the size of the L1 insertion and on the zygosity of the filled allele. The second reaction amplified the 3' L1-genomic junction using a primer in the flank (identical to one of those used to amplify the full locus) and another one anchored in the L1 3' end (RB3PA1). Finally, a similar reaction was also performed for the 5' junction for putative full-length insertions using the RB5PA1 primer anchored in the L1 5' end. PCR amplification was performed in 50-µL reactions containing 20 ng of genomic DNA, 1X PCR Buffer, 0.2 µM of primers, 0.2 µM dNTPs, 1.5 mM MgCl$_2$, and 1 U of Platinum Taq DNA Polymerase (Life Technologies). Amplification was performed with an Applied Biosystems Veriti Thermal Cycler (Life Technologies) under the following cycling conditions: one step at 94°C for 2 min, one step of 35 cycles at 94C for 30 s, 56C for 30 s, 72°C for 1 min, and one final step at 72°C for 7 min. PCR products were resolved and visualized by 1.5% agarose gel electrophoresis in 0.5X TBE buffer and ethidium bromide staining (50 ng/mL, Life Technologies). When amplification of the entire locus spanning the L1 insertion site failed (neither empty nor filled locus amplified, or non-specific banding pattern), we interpreted the result as a PCR failure. When the latter gave a specific product, if the 3' flank and/or the 5' flank PCR reaction gave a predominant unique product of the expected size, the insertion was considered as validated, and if not, as non-validated. Overall the rate of true positive was 94%.

## Detection of 3' transductions in ATLAS-seq sequencing data

Demultiplexed 3' ATLAS-seq reads, trimmed for the barcode and ATLAS-linker were successively trimmed at their 3' end using cutadapt to remove the RB3PA1 ATLAS primer, the L1 sequence and a polyT stretch. Parameters were identical to regular ATLAS-seq read processing, except for polyT trimming cutadapt options (-a T{100} which ensures only trimming of the most distal poly(T)). We mapped reads using bwa mem (-M -t 4 options), which allows soft clipping, and samtools view with awk scripts to extracts reads with a CIGAR string indicative of 3' end clipping, and to obtain the genomic location and mapping quality of the clipped part of the read (SA tag). At this stage we only kept read for which both split fragments were able to map unambiguously to the reference genome (MAPQ>15). This filtering step is essential since the clipped region, potentially corresponding to a 3' transduced sequence, is often of low complexity. BEDtools window was next used to identify among the split reads, those for which the 5' end mapped next to any L1HS-Ta sequence identified by ATLAS-seq and the 3' end next to a full-length L1HS-Ta instance also identified by ATLAS-seq, but distinct from the first one. Progenies with multiple potential progenitors were eliminated. Results of this analysis are detailed in *Supplementary file 3* and summarized in *Figure 5a* and *Supplementary file 5*.

## L1 lineage identification

This section is related to *Supplementary file 5*. Some L1 elements with high retrotransposition activity ('hot' L1) belong to well-defined lineages with distinctive 3' transductions. Three such main lineages have been extensively characterized (AC002980, RP and LRE3) (*Schwahn et al., 1998*; *Brouha et al., 2002*; *Myers et al., 2002*; *Beck et al., 2010*; *Macfarlane et al., 2013*). We used a computational approach to identify among the full-length insertions found by ATLAS-seq those that belong to these lineages. First, we pooled all ATLAS-seq reads from all cell lines after barcode and linker clipping (but still containing the poly(A), L1 and flanking sequence). Then, for each read, we serially searched lineage-specific tags (see below) using cutadapt (parameters: -e 0.1 -O 16 -m 25 –no-trim –discard-untrimmed) (*Martin, 2011*). The reads containing one of the tags were mapped on the human reference genome using bwa mem (-M option), and filtered for unique mappers and primary alignments using samtools view (-q 15 –F 256 options) (*Li et al., 2009*; *Li and Durbin, 2010*). Finally, mapped reads intersecting full-length insertions were counted using bedtools intersect (-c option) (*Quinlan and Hall, 2010*). Since all three lineages belong to the most recent L1 subfamily (L1HS-Ta1d), we discarded matches corresponding to reference insertions of older subfamilies, and manually inspected the remaining insertions and their supporting reads. These last steps were necessary to eliminate a number of false positives for

the RP lineage, which is characterized by a very low sequence complexity and stretches of homo-polymers. At the end of this procedure, L1 copies with reads supporting a particular lineage were annotated as belonging to this lineage. This strategy allowed us to overcome the limitations due to reduced sequence quality within or downstream homopolymers and to identify informative reads spanning putative transductions. The sequences of the lineage-specific tags were as follow: AC002980, GCTTTATTGAGGTGTAACCAGCA; RP, TAAATTTAAAACTTTTTTTTTT; LRE3, CGGAA TAGACATTTTGCTTTTCT. They are identical - or shortened - reverse-complemented version of previously published oligonucleotide sequences used to amplify distinctive L1 lineages (TS-ATLAS) (*Macfarlane et al., 2013*).

To verify the sensitivity of this approach, we compared our list of full length insertions with a published dataset of insertions deriving from the same three lineages (AC002980, RP and LRE3) (*Macfarlane et al., 2013*). Out of 15 full-length insertions previously identified by MacFarlane *et al.*, 8 are present in the cell lines studied here, and all 8 were identified by our computational approach (100% recovery). However, none was highly expressed in any of the cell types of our panel. In addition, we identified 2 additional copies with a 3' transduction typical from the RP lineage. In total, only 1 element out of 10, inserted in the *NEDD4* gene (EXP_ID_0447), was ranked among the top 20 most highly expressed L1 copies. For this particular copy, we confirmed the presence of the typical RP transduction by Sanger sequencing. Consistent with its affiliation to the 'hot' L1 RP lineage, this copy is also retrotransposition-competent (*Figure 5b*).

## Cloning of full length L1-HS

Cloning of L1 copy EXP_ID_0447 was obtained by nested PCR amplification from 50 ng of MCF7 genomic DNA using Phusion High-Fidelity DNA Polymerase (Thermo Scientific, Waltham, MA) in 50-µL reactions. The first PCR was performed with primers specific to the genomic DNA sequence flanking L1 copy EXP_ID_0447 (LOU1652 and LOU1656) with the following program: 98°C for 3 min, 35 cycles [98°C for 10 s, 54°C for 15 s, 72°C for 3.5 min], and 72°C for 10 min. A ~6 kb PCR product was resolved by 0.8% agarose gel electrophoresis, gel-purified using Wizard SV Gel and PCR Clean-Up System (Promega), and resuspended in 50 µL of milli-Q water. The second PCR was performed using 1 µL of the purified PCR product with primers matching the L1HS 5' UTR (LOU1662) and the downstream flanking genomic DNA sequence of the L1 copy EXP_ID_0447 (LOU1664) with the following program: 98°C for 3 min, 40 cycles [98°C for 10 s, 58°C for 15 s, 72°C for 4 min], and 72°C for 10 min. A PCR product of ~6 kb fragment was gel-purified as described above. Both primers of the second PCR contained homology in their 5' parts to the CMV promoter and to the retrotransposition reporter cassette contained in the L1 expression vector pAD135 (*Doucet et al., 2010*), allowing SLiCE cloning, a method based on in vitro homologous recombination (*Zhang et al., 2012*). Briefly, pAD135 was digested with NotI and BstZ17I to remove the engineered L1.3 element from the expression vector. SLiCE cloning to generate pVan609 plasmids was conducted in a 10-µL reaction by combining 100 ng of the digested and gel-purified vector fragment with the PCR product of L1 EXP_ID_0447 in a 1:3 molar ratio, 1X SLiCE buffer and 1 µL of PPY SLiCE extract for 1 hr at 37C (*Zhang et al., 2012*). Sanger sequencing of positive clones confirmed the specific subcloning of the full-length L1 copy EXP_ID_0447 through the identification of the 3' flanking region of genomic DNA. The flanking genomic DNA sequence was then removed by digestion of pVan609 clones with BstZ17I, an enzyme that cuts in the 3' UTR of the subcloned L1 and just upstream of the retrotransposition reporter in the vector. This strategy allows, after re-ligation of the vector to itself to generate pVan610 clones, containing a full length L1 copy with an engineered 3' UTR containing a retro-transposition reporter. Four independent pVan610 clones (pVan610-1 to -4) were generated to rule out mutations that could have occurred through the second round of PCR amplification.

## Cellular retrotransposition assay

L1 retrotransposition assay was conducted as described previously (*Moran et al., 1996*; *Wei et al., 2000*) with minor modifications. Briefly, HeLa cells were plated in 6-well plates at $2 \times 10^5$ cells per well. The next day, cells were transfected with 1 µg of plasmid DNA and 3 µL of Lipofectamine 2000 (Life Technologies) diluted in 200 µL of Opti-MEM (Life Technologies), and medium was replaced with fresh medium after ~5 hr. Two days after transfection, medium was

supplemented with G418 (Life Technologies) at 1 mg/mL. After 10 days of selection, colonies were fixed and stained with 1 mL of a solution containing 30% methanol (v/v), 10% acetic acid (v/v) and 0.2% Coomassie blue (m/v). The colonies obtained in three independent biological replicates were manually counted to assess the retrotransposition efficiency of pVan610 clones compared to the positive control pJM101/L1.3 (*Sassaman et al., 1997*) and negative control pJM105/L1.3 (RT mutant) (*Wei et al., 2001*). For each biological replicate (four independent L1 clones, independently transfected three times each), two wells were used as internal technical replicates (parallel transfections).

## shRNA-mediated L1 RNA knockdown

Annealed complementary oligonucleotides with 3' overhangs (see *Supplementary file 2* for sequences) were directly cloned into pLKO.1 Puro lentiviral vectors (Sigma-Aldrich, St Louis, MO) between AgeI and EcoRI sites. Lentiviral particles were prepared by phosphate-calcium-mediated transfection of the vector and helper plasmids into HEK-293T cells. Briefly, one day before transfection, we plated $3.5 \times 10^6$ cells per 10 cm-dish. Cells were transfected with 8.6 µg of shRNA pLKO.1 construct, 8.6 µg of pCMV-dR8.91 (Addgene #2221) and 2.8 µg of pCMV-VSV-G (Addgene #8454). Growth medium was changed 8 hr post-transfection. Two days after transfection, cells supernatants from 5 dishes were collected, pooled, filtered, and concentrated by overnight centrifugation at 4,000 rpm. Pellets were resuspended in 350 µL of growth medium, aliquoted and stored at -80°C until use. Viral stocks were titrated by colony formation assay upon puromycin selection (8 days, 1 µg/mL) using serial dilution of vector stocks. For RNA-seq experiments, we infected 500,000 cells in triplicates (either MCF7 or 2102Ep cells) at M.O.I 1. Cells were collected for protein and RNA extraction after 1 week of puromycin selection (1.5 µg/mL).

## RNA-sequencing (RNA-seq)

Whole cell total RNA was processed by Beckman Coulter Genomics according to the following steps. Poly(A)+ RNA was isolated from the total RNA samples and fragmented with ultrasound (1 pulse of 15 s at 4°C). First-strand cDNA synthesis was primed with random hexamers. Then, the Illumina TruSeq sequencing adapters were ligated to the 5' and 3' ends of the cDNA. The cDNA was finally amplified with PCR using a proof reading enzyme and between 12 and 14 cycles. The TruSeq barcode sequences which are part of the 3' TruSeq sequencing adapters are included in *Supplementary file 2*. The cDNA was purified using the Agencourt AMPure XP kit (Beckman Coulter). For Illumina sequencing, the cDNA samples were pooled in approximately equimolar amounts. The cDNA pool was size fractionated in the size range of 350–550 bp on a preparative agarose gel and size range was verified by capillary electrophoresis (Shimadzu MultiNA microchip electrophoresis system). Finally, 2x150 bp paired-end and strand-specific sequencing was performed with HiSeq SBS kit v4 on a HiSeq 2500 system (Illumina, San Diego, CA). RNA-seq sequencing statistics are summarized in *Supplementary file 2*.

## Data

Raw ATLAS-seq and RNA-seq data were submitted to the ArrayExpress database (www.ebi.ac.uk/arrayexpress) under accession number E-MTAB-4676 and E-MTAB-3788, respectively. RNA-seq data contain biological duplicates for untreated cells (two independent RNA extractions), and triplicates for shRNA-treated cells (three parallel lentivirus shRNA transductions, see above). The genomic locations of L1 insertions in the different cell lines are provided in *Supplementary file 3*. Published RNA-seq data used in this work are described in *Supplementary file 1*, and ChIP-seq data were obtained from the ENCODE Project: H3K9me3 (ENCSR000EWQ), H3K4me3 (ENCSR000DWJ), H3K27Ac (ENCSR000EWR), H3K27me3 (ENCSR000EWP), and Pol2 (ENCSR000DMT). All RNA-seq and ChIP-seq datasets contain biological duplicates except HCT 116 RNA-seq (single experiment). Copy number variation (CNV) data were obtained from the ENCODE Project: BJ (ENCFF628TJX), H1 (ENCFF228MUH), HeLaS3 (ENCFF996ASY), HepG2 (ENCFF074XLG), IMR90 (ENCFF455ARY), K562 (ENCFF486MJU) and MCF7 (ENCFF278UJF).

## Pre-processing of RNA-seq libraries

RNA-seq raw reads were pre-processed to enhance their quality and to avoid a significant loss at the mapping stage. The quality of raw reads was verified for each lane per RNA-seq sample, using the FASTQC (v0.11.2) tool (http://www.bioinformatics.babraham.ac.uk/projects/fastqc/). Reads were trimmed using sliding-window mode from Trimmomatic (v0.32) (*Bolger et al., 2014*). The minimum quality and read length allowed were 20 and 25, respectively, as they showed the best mapping rates in several tests trying different values of these parameters. The rest of the Trimmomatic parameters were set as recommended for paired-end (PE) reads in the Trimmomatic tool manual.

## RNA-seq mapping

PE trimmed reads were aligned per lane to both the hg19 reference transcriptome and genome obtained from UCSC, using TopHat2 v2.0.13 (*Kim et al., 2013*). For a better accuracy, the parameter –read-realign-edit-dist was set to 0 (specifying alignment of all reads at each step, returning the best alignment for each read). The –r (insert size) and –mate-std-dev (insert size standard deviation) parameters were estimated using the tool CollectInsertSizeMetrics from Picardtools v1.128 (http://broadinstitute.github.io/picard), after mapping a subset of the RNA-seq samples. The other mapping parameters were set to their default values.

## Comparison of global L1HS-Ta RNA levels between cell lines

This section is related to *Figure 1b*. Reads were mapped against the consensus L1HS sequence from RepBase (*Bao et al., 2015*) using the end-to-end mode from bowtie2 (v2.2.4) (*Langmead and Salzberg, 2012*). The total end-to-end mismatches (specified by -score-min) were -0.6 + 0.6*L, so -45.6 for 75 bp reads (or no more than 7 high-quality mismatches) or -90.6 for 150 bp reads (or no more than 15 high-quality mismatches). These options allow two mismatches per seed, needed to allow the SNPs that distinguish each L1HS subfamilies. Reads that mapped to a diagnostic tri-nucleotide (5927–5929) within the L1 sequence were extracted using samtools (*Li et al., 2009*), and the occurrences of each tri-nucleotide at this position was counted. These counts of tri-nucleotide-containing reads per library were normalized by the total number of reads mapped to the hg19 reference genome. Reads with an ACA diagnostic tri-nucleotide were counted as derived from L1HS-Ta elements.

## Transcriptional analysis of individual L1HS copies

This section is related to *Figure 3e* and *Figure 4—figure supplement 1*. Stranded coverage file (wig files) were generated from the alignments with the tool 'count' from igvtools (v2.3.43) (http://www.broadinstitute.org/igv). Only the strand of first mates and with a MAPQ of 20 were taken into account, ensuring that only one read per fragment was counted and only pairs uniquely mapped were retained for downstream analyses. The window size was set to 10 and the window function used was the median. Coverage wig files from all libraries were normalized according to the total number of mapped mates 1 of each library. After normalization, wig files were converted to bigWig files using the command wigToBigWig (v4) from UCSC bioinformatics utilities. The bigWig files were used to identify transcriptional signals from the possibly full-length L1HS individual copies: RNA-seq signals upstream or downstream individual L1HS copies were used as a proxy to identify potential transcriptional signature of specific individual L1HS copies, as these genomic flanking sequences are more-often uniquely mappable. To obtain a global view of the transcription levels detectable at each LINE-1 locus, the computeMatrix command from deeptools (v1.5.9.1) was employed (*Ramírez et al., 2014*). For each RNA-seq strand and L1HS strand, the scores associated at the 5' and 3' of the L1HS copies in a 1 kb window size, upstream and downstream respectively, were computed separately and averaged between biological replicates. The window size was selected according to previous studies and to avoid overlapping of transcription signals from other genomic features. Annotated exons within the 1 kb window were masked. Heatmaps were produced with the R package heatmap.2, and lines were sorted according to the total transcription signal at the downstream flanking region.

## Analysis of ChIP-seq datasets

This section is related to *Figure 3f*. Mapped ChIP-seq data were retrieved from ENCODE. The replicates of each experiment were pooled together and read duplicates were removed from the merged bam files. The merged bam files were converted into wig files using 'count' from igvtools, with a window of 10 bp and the median function, and, then, converted into bigWig file with tool wigToBigWig from the UCSC tools. ChIP-seq signals were obtained using the 'reference-point' mode of computeMatrix from deeptools, specifying 1 kb upstream and 1 kb downstream of the TSS & TES, which were defined as the extremities of each L1HS copy, and with a window of 50bp and mean function. Using a threshold of 0.05 on the FPKM L1HS expression values, the L1HS copies were divided in two groups: expressed and non-expressed copies. The mean of each group of ChIP-seq signals was extracted and plotted.

## Effect of ORF1 shRNA on the expression of individual L1 copy

This section is related to *Figure 3—figure supplement 2*. Separate annotation files (.gtf) were created for each cell line by appending the coordinates of 1 kb downstream of each L1HS copy to the hg19 annotation file. To count the number of reads associated with genomic features (exons and flanking L1 sequences), HTSeq v0.6.1p1 was employed in the union mode, and a MAPQ threshold of 50 was used to take only unambiguously mapped reads into account (*Anders et al., 2015*). Other parameters were assigned to their default value. In order to determine whether RNA-seq tags mapping within the 1 kb downstream of each L1HS actually reflect L1HS-derived transcripts, a differential expression analysis (DEA) was performed using HTSeq counts as inputs. Since reads mapping within the downstream 1 kb sequence flanking L1HS instances are often non-abundant, several DEA tools were tested for their ability to discriminate properly differential expression with low number of counts: from this test, DESeq2 (*Love et al., 2014*) gave the best performance (data not shown) and was used for the rest of the analysis. Data corresponding to each L1 shRNA was compared against control data corresponding to a scrambled shRNA, and default parameters were used for the entire analysis.

## Clustering of individual L1-HSTa loci according to their transcriptional activity

This section is related to *Figure 4a*. The transcriptional activity of each L1-HSTa instance was estimated independently for each cell-line in which it is detected by the number of RNA-seq fragments which non-redundantly align to the downstream adjacent 1 kb of genomic sequence, and normalized to the number of non-redundantly aligned reads in the entire RNA-seq dataset (FPKM). For this purpose, mates 1 stranded-specific counting was performed at 1 kb downstream L1HS copies with a dedicated script. Only mates with a MAPQ>= 20 were taken into account. The FPKM for each copy was calculated according to the total number of mapped mates 1 in each library. Clusters of cell-lines with similar patterns of detectable expression of distinct L1 copies were clustered using the R hclust function (distance measure='euclidean', clustering method='ward'). To identify groups of L1s which exhibit similar behaviour across cell-types, L1 expression was compared between cell-types using the reciprocal of the mean product of FPKMs as a distance measure. L1s with no detectable expression in any cell-type were excluded, and pairs of L1s which are not present in any common cell-types were assigned a neutral distance equal to the mean of all calculable pairwise distances. Clustering was performed using R hclust (clustering method='ward').

## RT-PCR validation of L1 chimeric transcripts

First-strand complementary DNAs (cDNA) were obtained from the RNA samples using SuperScript III Reverse Transcriptase (Life Technologies) as per the manufacturer's instructions. Briefly, 1 μg of DNase-treated RNA samples were combined in a 13-μL reaction including 1 μL of a 10 mM dNTP mix, 1 μL of a 50 μM RACE primer and milli-Q water. The samples were incubated for 5 min at 65C and placed on ice for 1 min. The reactions were then supplemented with a 7-μL cDNA Synthesis Mix containing 4 μL of 5X First-Strand buffer, 1 μL of 0.1 M DTT, 1 μL of 40 U/μL RNaseOUT and 1 μL of 200 U/μL of SuperScript III RT. For each sample, a control condition was prepared without reverse transcriptase. The tubes were incubated for 50 min at 55°C and then for 15 min at 70°C. PCR amplification of the cDNA was obtained using Platinum Taq DNA polymerase (Life Technologies) as per

the manufacturer's instructions. Briefly, 1 μL of cDNA sample was combined in a 50-μL reaction with 1X PCR Buffer, 1.5 mM MgCl$_2$, 1 U of Platinum Taq DNA polymerase, 0.2 mM dNTP, 0.2 μM of forward primer (RB3PA1 [*Badge et al., 2003*]), 0.2 μM of reverse primer (specific to the 3' flanking genomic DNA sequence of L1 copies, see *Supplementary file 2* for details). Amplification was performed with an Applied Biosystems Veriti Thermal Cycler (Life Technologies) with the following program: one step at 94°C for 2 min, one step of 32 cycles at 94°C for 30 s, 56°C for 30 s, 72°C for 1 min, and one final step at 72°C for 5 min. The detection of GAPDH transcripts was performed similarly, with only 30 cycles of PCR amplification, using primers LOU0639 and LOU0576. PCR products were resolved and visualized by 1.5% agarose gel electrophoresis in 0.5X TBE buffer and ethidium bromide staining (50 ng/mL, Life Technologies).

## Relationship of expressed copies with genes and copy number variants

This section is related to *Figure 4c–d* and *Figure 4—figure supplement 3*. Raw RNA-seq data retrieved from ENCODE were processed as described above to obtain an FPKM value for each gene. We used the UCSC gene set (hg19 version) obtained from iGenomes (https://support.illumina. com/sequencing/sequencing_software/igenome.html). When an L1 copy was inserted in a region corresponding to several overlapping genes oriented in the same direction, we considered the most expressed gene. Ploidy at each L1 insertion locus and for each cell line analyzed was determined using the copy number variation (CNV) tracks obtained from ENCODE (see 'Data' section). They were deduced from Illumina Human 1M-Duo Infinium HD BeadChip assays and circular binary segmentation (CBS), which segments the genome into amplified, normal, heterozygous deletion, homozygous deletion or uncovered regions. As expected, no L1 were inserted in region annotated as homozygous deletion. Uncovered regions were excluded from the analysis. The list of full-length L1 insertions provided in *Supplementary file 5* was divided in two groups, expressed and non-expressed copies, using a threshold of 0.05 on the FPKM L1HS expression values. Due to CNV data availability, only L1 elements identified in BJ, H1, HeLaS3, HepG2, IMR90, K562 and MCF7 were considered for this analysis.

## Acknowledgements

We thank the IRCAN genomics core facility and C Baudoin for Ion Torrent sequencing. We are grateful to the ENCODE Consortium and particularly to the laboratories of T Gingeras (Cold Spring Harbor Laboratory, USA), of J Stamatoyannopoulos (Univ. of Washington, USA), of V. Iyer (Univ. of Texas at Austin, USA), and of P Farnham (Univ. of Southern California, USA) for generating RNA-seq and ChIP-seq data, to the laboratory of R Myers (Hudson Alpha Institute for Biotechnology, USA) for generating CNV data, to PW Andrews (Univ. of Sheffield, UK) for providing the 2102Ep cell line, to J-L Garcia-Perez (Genyo, Univ. of Granada, Spain) for providing H1 genomic DNA, to JV Moran (Univ. of Michigan, USA) for providing control plasmids for retrotransposition assay, and to R Tirado-Magallanes (ENS, Paris, France) for help with the ChIP-seq aggregation plots. This work was supported by grants to GC from the Fondation ARC pour la recherche sur le cancer (Projet Fondation ARC, 20141201838), the Cancéropôle PACA (AAP Emergence), the European Research Council (ERC-2010-StG 243312, Retrogenomics), the French Government (National Research Agency, ANR) through the 'Investments for the Future' (LABEX SIGNALIFE, #ANR-11-LABX-0028-01), the Fondation pour la Recherche Médicale (FRM DEP20131128533), and CNRS (GDR 3546). MK was supported by a PhD fellowship from the Ligue Nationale Contre le Cancer. DVL. was supported by a COTACyT-CONACyT international postgraduate fellowship.

## Additional information

### Funding

| Funder | Grant reference number | Author |
|---|---|---|
| Fondation ARC pour la Recherche sur le Cancer | Projet Fondation ARC, 20141201838 | Gaël Cristofari |
| Fondation pour la Recherche Médicale | DEP20131128533 | Gaël Cristofari |

| Canceropôle PACA | AAP Emergence | Gaël Cristofari |
| Agence Nationale de la Recherche | Labex SIGNALIFE, ANR-11-LABX-0028-01 | Gaël Cristofari |
| Ligue Contre le Cancer | PhD fellowship | Monika Kuciak |
| European Research Council | Starting Grant, Retrogenomics, 243312 | Gaël Cristofari |
| Consejo Nacional de Ciencia y Tecnología | COTACyT-CONACyT, International Postgraduate Fellowship | Dulce B Vargas-Landin |
| Centre National de la Recherche Scientifique | GDR 3546 | Gaël Cristofari |

The funders had no role in study design, data collection and interpretation, or the decision to submit the work for publication.

### Author contributions

CP, established the ATLAS-seq procedure, designed, performed and analyzed the biological experiments, and revised the final manuscript; DBV-L, designed, performed and analyzed the biological experiments, designed and conducted the computational analyses, and revised the final manuscript; AJD, designed, performed and analyzed the biological experiments, and revised the final manuscript; DvE, designed and conducted the computational analyses, and revised the final manuscript; JV-O, PN, designed, performed and analyzed the biological experiments; MK, established the ATLAS-seq procedure; AC, contributed to early analyses of ATLAS-seq data; GC, supervised the project, designed and conducted the computational analyses, and drafted the manuscript

### Author ORCIDs

Dulce B Vargas-Landin, http://orcid.org/0000-0002-4773-9406
Aurélien J Doucet, http://orcid.org/0000-0001-7221-4874
Jorge Vera-Otarola, http://orcid.org/0000-0003-2547-4213
Gaël Cristofari, http://orcid.org/0000-0001-5620-3091

## Additional files

### Supplementary files

• Supplementary file 1. Description of the cell lines and RNA-seq datasets used in this study. Note that HEK-293T data were ambiguously named in the original publication, as 'HEK-293' in the main text, but as 'HEK-293T' in the method section (*Sultan et al., 2014*). We solved this ambiguity by searching for RNA-seq reads matching the SV40 virus and Neomycin-resistance gene sequences, which confirmed the nature of the cells as being 'HEK-293T'.

• Supplementary file 2. Oligonucleotides used in this study, RNA-seq and ATLAS-seq statistics. Note that primers used to PCR-validate L1 insertions are described in *Supplementary file 3* along with the PCR results.

• Supplementary file 3. Coordinates of all L1HS-Ta elements mapped in a panel of 12 human cell lines, evidence for 3' transductions, and correspondence between insertion IDs. See first sheet for legend.

• Supplementary file 4. ATLAS-seq PCR validation results obtained in HEK-293T cells.

• Supplementary file 5. Levels of expression, retrotransposition capability and lineages of the full length L1HS-Ta copies mapped by ATLAS-seq. Values were used to construct the heat map shown in *Figure 4a* and the pie chart in *Figure 5a*.

## Major datasets

The following datasets were generated:

| Author(s) | Year | Dataset title | Dataset URL | Database, license, and accessibility information |
|---|---|---|---|---|
| Philippe C, Vargas-Landin DB, Doucet AJ, van Essen D, Vera-Otarola J, Kuciak M, Corbin A, Nigumann P, Cristofari G | 2016 | A comprehensive map of L1HS-Ta retrotransposons in a panel of human cells obtained by ATLAS-seq | http://www.ebi.ac.uk/ar-rayexpress/experiments/E-MTAB-4676 | Publicly available at the EBI European Nucleotide Archive (accession no: E-MTAB-4676) |
| Philippe C, Vargas-Landin DB, Doucet AJ, van Essen D, Vera-Otarola J, Kuciak M, Corbin A, Nigumann P, Cristofari G | 2015 | RNA-seq of MCF-7 (breast adenocarcinoma) and 2102Ep (embryonic carcinoma) cells upon LINE-1 knockdown | http://www.ebi.ac.uk/ar-rayexpress/experiments/E-MTAB-3788/ | Publicly available at the EBI European Nucleotide Archive (accession no: E-MTAB-3788) |

The following previously published datasets were used:

| Author(s) | Year | Dataset title | Dataset URL | Database, license, and accessibility information |
|---|---|---|---|---|
| Gingeras T | 2011 | HeLa S3 RNA-seq | https://www.encodeproject.org/experiments/ENCSR000CPR/ | Publicly available at ENCODE (accession no. ENCSR000CPR) |
| Gingeras T | 2011 | MCF-7 RNA-seq | https://www.encodeproject.org/experiments/ENCSR000CPT/ | Publicly available at ENCODE (accession no. ENCSR000CPT) |
| Gingeras T | 2011 | HepG2 RNA-seq | https://www.encodeproject.org/experiments/ENCSR000CPE/ | Publicly available at ENCODE (accession no. ENCSR000CPE) |
| Gingeras T | 2012 | IMR-90 RNA-seq | https://www.encodeproject.org/experiments/ENCSR000CTQ/ | Publicly available at ENCODE (accession no. ENCSR000CTQ) |
| Gingeras T | 2011 | BJ RNA-seq | https://www.encodeproject.org/experiments/ENCSR000COP/ | Publicly available at ENCODE (accession no. ENCSR000COP) |
| Sultan M, Amstislavskiy V, Risch T, Schuette M, Dökel S, Ralser M, Balzereit D, Lehrach H, Yaspo ML | 2014 | Influence of RNA extraction methods and library selection schemes on RNA-seq data. | https://www.ebi.ac.uk/ena/data/view/ERP003460 | Publicly available at the EBI European Nucleotide Archive (accession no: ERP003460) |
| Sánchez Y, Segura V, Marín-Béjar O, Athie A, Marchese FP, González J, Bujanda L, Guo S, Matheu A, Huarte M | 2014 | Genomewide analysis of the human p53 transcriptional network unveils a lncRNA tumor suppressor signature | http://www.ebi.ac.uk/ena/data/view/SRP043273 | Publicly available at the EBI European Nucleotide Archive (accession on: SRP043273) |
| Gingeras T | 2011 | K562 RNA-seq | https://www.encodeproject.org/experiments/ENCSR000CPH/ | Publicly available at ENCODE (accession no. ENCSR000CPH) |
| Gingeras T | 2011 | H1 RNA-seq | https://www.encodeproject.org/experiments/ENCSR000COU/ | Publicly available at ENCODE (accession no. ENCSR000COU) |

| | | | | |
|---|---|---|---|---|
| Farnham P | 2012 | MCF-7 H3K9me3 ChIP-seq | https://www.encodeproject.org/experiments/ENCSR000EWQ/ | Publicly available at ENCODE (accession no. ENCSR000EWQ) |
| Stamatoyannopoulos J | 2011 | MCF-7 H3K4me3 ChIP-seq | https://www.encodeproject.org/experiments/ENCSR000DWJ/ | Publicly available at ENCODE (accession no. ENCSR000DWJ) |
| Farnham P | 2012 | MCF-7 H3K27ac ChIP-seq | https://www.encodeproject.org/experiments/ENCSR000EWR/ | Publicly available at ENCODE (accession no. ENCSR000EWR) |
| Farnham P | 2012 | MCF-7 H3K27me3 ChIP-seq | https://www.encodeproject.org/experiments/ENCSR000EWP/ | Publicly available at ENCODE (accession no. ENCSR000EWP) |
| Ayer V | 2011 | MCF-7 Pol2 ChIP-seq | https://www.encodeproject.org/experiments/ENCSR000DMT/ | Publicly available at ENCODE (accession no. ENCSR000DMT) |
| Myers R | 2011 | HudsonAlpha human BJ genotype | https://www.encodeproject.org/experiments/ENCSR000BAG/ | Publicly available at ENCODE (accession no. ENCFF628TJX) |
| Myers R | 2011 | HudsonAlpha human H1-hESC genotype | https://www.encodeproject.org/experiments/ENCSR000BAO/ | Publicly available at ENCODE (accession no. ENCFF228MUH) |
| Myers R | 2011 | HudsonAlpha human HeLa-S3 genotype | https://www.encodeproject.org/experiments/ENCSR000BBE/ | Publicly available at ENCODE (accession no. ENCFF996ASY) |
| Myers R | 2011 | HudsonAlpha human HepG2 genotype | https://www.encodeproject.org/experiments/ENCSR000BBN/ | Publicly available at ENCODE (accession no. ENCFF074XLG) |
| Myers R | 2011 | HudsonAlpha human IMR90 genotype | https://www.encodeproject.org/experiments/ENCSR000AZE/ | Publicly available at ENCODE (accession no. ENCFF455ARY) |
| Myers R | 2011 | HudsonAlpha human K562 genotype | https://www.encodeproject.org/experiments/ENCSR000BBO/ | Publicly available at ENCODE (accession no. ENCFF486MJU) |
| Myers R | 2011 | HudsonAlpha human MCF7 genotype | https://www.encodeproject.org/experiments/ENCSR000AZH/ | Publicly available at ENCODE (accession no. ENCFF278UJF) |

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
