## [Decision Letter]

Thank you for submitting your work entitled "Activation of individual L1 retrotransposon instances is restricted to cell-type dependent 'hot loci'" for consideration by *eLife*. Your article has been reviewed by two peer reviewers, and the evaluation has been overseen by a guest Reviewing Editor, who acted as one of the reviewers, and Diethard Tautz as the Senior Editor.

The reviewers agree that your manuscript provides an important contribution to the field and will be of interest to *eLife* readers. They also recommend some revisions as outlined below.

The manuscript addresses a long-standing question in our field: Namely, how many and which of the many genomic LINE-1 loci contribute to expression at the RNA level. The authors do a very nice job of integrating different types of data: RNA-seq, epigenetic studies, and LINE-1 mapping with ATLAS, to reach conclusions about this. They take into account sense/antisense orientations and leverage that information in order to 'see' reads that span the L1 3' and 5' junctions with the adjacent genomic DNA. The attention to detail is high, and I think their conclusion – that a restricted subset of L1HS-Ta loci, a subset of which is polymorphic among individual cell lines, is responsible for the bulk of L1 expression – is sound.

The following major points need to be addressed in a revision, before the paper can be accepted:

More information should be provided on the positions of these active LINE-1 / contrast them to inactive loci. What are the surrounding genes and the relationship between the gene loci and the LINE-1? What is the ploidy/copy-number at these sites?

More discussion is required on how the authors think their shRNA to ORF1 works. Their language ("Impact of shRNA-mediated ORF1 knockdown on the transcription of all L1HS genomic instances") suggests they believe a shRNA silencing effect at the level of the genome is at work. One would think that shRNA is just causing degradation of the mRNA, which is what accounts for reductions in the reads.

'Hot' is a colloquialism for retrotransposition active LINE-1. We would suggest the authors think of different terms, for example, expressed LINE-1 in place of hot LINE-1 when RNA levels are described, permissive loci in place of hot loci when considering epigenetic context. Keeping the narrow definition of 'hot' may avoid confusion in the literature.

You should consider utilising the identification of daughter elements from their shared 3' transductions with putative progenitors as evidence of retrotransposition capability (where this was not assessed directly by cell culture assays). However you do not comment on the relationship between these transduction lineages and previously identified highly active pedigrees of elements. It is well established that common, highly active polymorphic L1 insertions, while showing significant allelic variation in retrotransposition activity, nevertheless frequently belong to lineages with distinctive 3' transductions. It would be valuable to know what proportion of the highly expressed elements analysed for their retrotransposition capability (either directly or by implication from ATLAS-seq reads) belong to previously identified lineages. It may be that the apparent dominance of small cohorts of elements is modulated by intrinsic features of these subsets of L1 elements.

---

## [Author Response]

*More information should be provided on the positions of these active LINE-1 / contrast them to inactive loci. What are the surrounding genes and the relationship between the gene loci and the LINE-1? What is the ploidy/copy-number at these sites?*

We have now compared the cohorts of expressed vs. non-expressed L1 copies with regards to:

i) Their proportion in genes;

ii) For the genic subset:

– Their orientation relative to genes;

– The levels of expression of their associated genes;

iii) Their proportion in copy number variation regions.

These analyses do not support a specific enrichment of expressed copies in genes or in amplified regions of the genome. For the genic copies, however, we found that expressed L1 copies are inserted in genes exhibiting higher levels of expression than inactive L1 loci.

In the revised manuscript, these results are now shown in Figure 4 and Figure 4—figure supplement 3 and described in an entirely new paragraph entitled “Relationship of expressed L1HS-Ta copies with genes and copy-number variants”:

“To test whether the expression of L1HS-Ta copies could be influenced by their genic environment, we first compared the proportion of expressed L1 insertions in genes as compared to non-expressed copies. […] The majority of L1 copies were inserted in normal regions (Figure 4—figure supplement 3), whether they were expressed or not, and expressed L1 copies were not significantly enriched in amplified regions.”

Related additional information has been added in the Materials and methods section, and we appended an additional column containing gene names to [Supplementary-material SD5-data].

*More discussion is required on how the authors think their shRNA to ORF1 works. Their language ("Impact of shRNA-mediated ORF1 knockdown on the transcription of all L1HS genomic instances") suggests they believe a shRNA silencing effect at the level of the genome is at work. One would think that shRNA is just causing degradation of the mRNA, which is what accounts for reductions in the reads.*

We agree that the title of Figure 3—figure supplement 2 was misleading. It has now been changed for “Impact of shRNA-mediated ORF1 knockdown on RNA levels for each L1HS-Ta genomic instance”.

*'Hot' is a colloquialism for retrotransposition active LINE-1. We would suggest the authors think of different terms, for example, expressed LINE-1 in place of hot LINE-1 when RNA levels are described, permissive loci in place of hot loci when considering epigenetic context. Keeping the narrow definition of 'hot' may avoid confusion in the literature.*

We agree that the use of ‘hot loci’ might be a source of confusion in the future. This was changed into ‘permissive loci’ in the title and throughout the manuscript, as appropriately proposed. Thus, the revised title is now: “Activation of individual L1 retrotransposon instances is restricted to cell-type dependent permissive loci”. (Please see answer 9. on the choice of ‘instances’ instead of ‘insertions’ and answer 6. on the revised Discussion).

*You should consider utilising the identification of daughter elements from their shared 3' transductions with putative progenitors as evidence of retrotransposition capability (where this was not assessed directly by cell culture assays). However you do not comment on the relationship between these transduction lineages and previously identified highly active pedigrees of elements. It is well established that common, highly active polymorphic L1 insertions, while showing significant allelic variation in retrotransposition activity, nevertheless frequently belong to lineages with distinctive 3' transductions. It would be valuable to know what proportion of the highly expressed elements analysed for their retrotransposition capability (either directly or by implication from ATLAS-seq reads) belong to previously identified lineages. It may be that the apparent dominance of small cohorts of elements is modulated by intrinsic features of these subsets of L1 elements.*

We addressed this question in two different ways:

i) First, we compared the list of full-length insertions, independently of their level of expression, with a published dataset of insertions deriving from the 3 main L1 lineages (RP, LRE3 and AC002980) (MacFarlane et al. 2013). Eight out of 15 full-length and lineage-specific insertions found by MacFarlane et al. are present in the studied cell lines, but none of them belong to the top 20 highly expressed L1 ([Supplementary-material SD5-data]).

ii) Second, we screened ATLAS-seq reads before poly(A) and L1 sequence trimming for the presence of lineage-specific sequence tags. This allowed us to identify the full-length L1 copies in our dataset belonging to these transduction lineages. Through this computational approach, we identified all 8 insertions previously found in the MacFarlane et al. dataset, as well as 2 additional insertions belonging to the RP lineage. One of these two insertions was ranked in the top 20 highly expressed L1s across all cell lines (L1N*_EDD4_*). Its affiliation to the RP lineage was confirmed by Sanger sequencing and its retrotransposition activity had already been tested by cellular retrotransposition in the initial version of the manuscript (now shown in Figure 5).

In conclusion, although our set of L1 insertions contains members of well-known transduction lineages, only one of them was ranked among the 20 top expressed L1 elements.

These results have now been incorporated as two additional columns in [Supplementary-material SD5-data], where they can be compared side-by-side with the expression levels and retrotransposition-capacity of each individual L1 insertion. In the main text, the results are described at the end of the paragraph entitled “Many transcribed L1HS-Ta are retrotransposition-competent”:

“Some L1 elements with high retrotransposition activity (‘hot’ L1) belong to well-defined lineages with distinctive 3’ transductions. […]Thus, these findings suggest that the observed high level of expression of a small cohort of L1 insertions is not an intrinsic feature of any previously identified lineage.”

Moreover, a new paragraph entitled “L1 lineage identification” has been added in the Materials and methods section.